## TECHNIQUES AND RESOURCES

# Developmental single-cell atlas of coronary vessel growth and cardiomyocyte interaction in zebrafish

Muhammad Abdul Rouf[1,2], Gülsüm Kayman Kürekçi[1,2,§], Shaoqiu Zhang[1,2,§], Stéphanie Larrivée Vanier[1,2,*], Sarah M. Kamel[1,2,‡], Ann Nee Lee[3], Ruey-Bing Yang[3], Shih-Lei Lai[3] and Rubén Marín-Juez[1,2,¶]

## ABSTRACT

Cardiac morphogenesis requires the intricate coordination of different cell types and molecular cues. Coronary vessel formation is essential for heart development, yet how coronary vessels grow and contribute to ventricular wall formation remains poorly understood. Here, we combine high-resolution imaging and new genetic tools to systematically analyze coronary vasculature development in zebrafish at each millimeter increment from 7 to 30 mm of body length and identify cellular and molecular milestones defining four distinct coronary network developmental stages. Our data show how coronary vessels expand, pattern, specify and act as vascular scaffolds to guide cardiomyocyte growth throughout development. Manipulating coronary network growth by Vegfa signaling perturbation through gain- and loss-of-function approaches disrupts coronary vessel–cardiomyocyte interactions and cardiomyocyte expansion. To gain further resolution into these processes, we build a single-cell RNA sequencing atlas by profiling 37,554 ventricular cells across the different developmental stages and identify new coronary markers, dynamic cellular interactions, and stage-specific endothelial–cardiomyocyte crosstalk. Overall, we present the first comprehensive roadmap of coronary vessel formation and coordinated cellular and molecular interactions with the developing cardiac muscle in zebrafish.

KEY WORDS: Coronary vasculature, Cardiomyocytes, Cardiovascular, Development, Vegfa, Zebrafish

## INTRODUCTION

The formation of a functional coronary vascular network is essential to support cardiac muscle growth. As cardiac morphogenesis progresses, coronary circulation arises to meet the increasing metabolic demands of the thickening ventricular wall (D'Amato et al., 2022; Red-Horse et al., 2010; Tian et al., 2015; Tomanek,

[1]Centre de Recherche Azrieli, Centre Hospitalier Universitaire Sainte-Justine, Montréal, QC, H3T 1C5, Canada. [2]Department of Pathology and Cell Biology, Faculty of Medicine, Université de Montréal, Montréal, QC, H3T 1J4, Canada. [3]Institute of Biomedical Sciences, Academia Sinica, Taipei 11529, Taiwan.
*Present address: Centre Hospitalier Universitaire de Sherbrooke Research Center, Université de Sherbrooke, Sherbrooke, QC, Canada. ‡Present address: Developmental, Stem Cell and Cancer Biology Program, The Hospital for Sick Children, Toronto, ON, Canada.
§These authors contributed equally to this work

¶Author for correspondence (ruben.marin.juez.hsj@ssss.gouv.qc.ca)

M.A.R., 0000-0002-4076-2291; R.M.-J., 0000-0001-5903-7463

2005). In mammals, coronary vessels develop during embryogenesis primarily from epicardium-derived cells (EPDCs), the endocardium and sinus venosus-derived endothelial cells (ECs) that invade the subepicardial and myocardial spaces, ultimately assembling into an intricate vascular network (D'Amato et al., 2022; Nakajima and Imanaka-Yoshida, 2013; Red-Horse et al., 2010; Riley, 2012; Wu et al., 2012). This process involves tightly coordinated interactions among coronary ECs, cardiomyocytes (CMs) and epicardial cells (Günthel et al., 2018; Gupta and Poss, 2012; Harrison et al., 2015; Marín-Juez et al., 2019; Pires-Gomes and Pérez-Pomares, 2013; Red-Horse et al., 2010; Wu et al., 2012).

Taking advantage of its strengths as a model organism and the rich repertoire of genetic tools available, recent studies have begun to investigate the mechanisms of coronary vessel formation in zebrafish (Bakis et al., 2023; Chiba et al., 2025; Duca et al., 2024; Gancz et al., 2019; Harrison et al., 2015; Karra et al., 2018; Lowe et al., 2019; Marín-Juez et al., 2019). At approximately 45 days post-fertilization (dpf), ECs from the atrioventricular canal (AVC) sprout and expand over the dorsal ventricular surface to form the coronary plexus (Chiba et al., 2025; Harrison et al., 2015). At the same stage, a subset of trabecular CMs breach the primordial CM layer to form the cortical myocardium (Gupta and Poss, 2012; Gupta et al., 2013). Cortical CMs follow the track of the developing coronary vasculature to populate the ventricular surface (Marín-Juez et al., 2019). A similar association between coronary ECs and CMs occurs during mouse heart development (DeBenedittis et al., 2022). Moreover, disruption of coronary development results in defective CM growth, mediated by impaired angiocrine signaling (Chiang et al., 2023; DeBenedittis et al., 2022; Rhee et al., 2018, 2021). These findings underscore the role of coronary vessels as active regulators of myocardial wall formation. However, the precise developmental dynamics of coronary vessel formation and interaction with CMs remain largely unknown.

Here, we combined high-resolution imaging, novel and established tissue-specific transgenic lines, functional manipulation of Vegfa signaling, and single-cell transcriptomics to construct a roadmap of coronary development. Spanning early emergence to maturation, we identify developmental milestones, define four distinct stages termed initiation, establishment, expansion and maturation, and show how the coronary network progressively populates the ventricular surface. These analyses reveal spatially coordinated proliferation between developing coronary ECs and adjacent CMs, supporting a vessel-guided model of myocardial expansion. Additionally, single-cell RNA sequencing (scRNA-seq) of 37,554 ventricular cells across developmental stages shows dynamic gene expression programs and cell type-specific interactions. Together, this integrative approach delineates the temporal and cellular architecture of coronary vascular development and identifies its instructive role in ventricular morphogenesis.

## RESULTS

### Coronary vasculature emergence and growth dynamics

To track coronary development, we used body length from the snout to the base of the tail as a proxy for developmental stage, as it more accurately reflects developmental progression than chronological age in zebrafish (Singleman and Holtzman, 2014). We used the *TgBAC(etv2:EGFP)* transgenic line to label coronary ECs and track coronary vessel emergence. When the body length was <8 mm, we observed no vessel-like structures on the heart (Fig. 1A,E). At this stage, *etv2*:EGFP expression was strong in the AVC, reported to be one of the origins of the coronary vasculature in zebrafish (Harrison et al., 2015) (Fig. 1A). As *etv2* is a transcription factor expressed in endothelial progenitors (Sumanas and Lin, 2006; van Bueren and Black, 2012), it is possible that this upregulation reflects the activation of vasculogenesis. At 8 mm of body length, the first coronary vessel was observed on the bulbus arteriosus (BA) (Fig. 1B,E), consistent with the extension of the hypobranchial artery onto the BA (Mizukami et al., 2023). At 9 mm, a second vessel appeared from the AVC (Fig. 1C,E), consistent with previous findings (Harrison et al., 2015). By 11 mm, both BA and AVC vessels continued to grow around the BA-ventricular base junction and the AVC, respectively (Fig. 1D,E). Therefore, between 8 and 11 mm of body length the first two coronary vessels emerge and expand over the BA and AVC regions. We refer to this period of coronary development as the initiation stage (Fig. S1A).

To assess vessel identity, we analyzed ventricles from *TgBAC(etv2:EGFP)*; *Tg(−0.8flt1:RFP)* fish. The combination of these transgenes identifies coronary ECs (EGFP[+]/RFP[+]) and coronary arteries (EGFP[+]/RFP[high]) (Harrison et al., 2015; Marín-Juez et al., 2016, 2019). Between 12 and 15 mm of body length, both the BA and the AVC vasculature continued to expand. At 12 mm, the BA sprout reached the ventricular base and upregulates −0.8flt1:RFP, indicating arterial identity (Fig. 1F). This vessel continued to expand over the left ventricular curvature to reach the apex (Fig. 1G-J). Based on its localization along the left ventricular curvature and arterial identity, we termed this vessel as the left coronary artery (LCA). The AVC-derived vasculature expanded over the dorsal ventricular surface and formed additional branches that extended dorsally and ventrally along the left curvature toward the apex, with some dorsal vessels becoming EGFP[+]/RFP[high] (Fig. 1F-J). We also observed a distinct dorsal EGFP[+]/RFP[low] vessel, potentially of AVC origin, which was located dorsal to and close to the LCA (Fig. 1I). This vessel and the LCA extended together along the left ventricular curvature toward the apex (Fig. 1I). Therefore, between 12 and 15 mm of body length, the dorsal vascular plexus is established, and the LCA grows to reach the apex. We refer to this period as the establishment stage (Fig. S1A).

As development progressed from 15 to 20 mm of body length, coronary vessels continued to grow and branch across both the dorsal and ventral ventricular surfaces (Fig. 1K-P, Fig. S2A-D). The coronary plexus continued to populate the dorsal surface and extended ventrally, consistent with observations in other organisms (Kattan et al., 1999; Red-Horse et al., 2010). From 16 mm onward, the coronary network expanded across the right ventricular curvature and sprouted toward the LCA, establishing a right-to-left expansion pattern over the ventral surface (Fig. 1L-N,Q), similar to observations in mice (Chen et al., 2014). When zebrafish reached 20 mm of body length, the expanding plexus contacted the LCA (Fig. 1N). To measure this ventral expansion, we quantified the number of branching points and the extent of coronary network coverage (Fig. 1O-Q). Our data show that both parameters increased significantly between 15 and 19 mm and remained unchanged

between 19 and 20 mm. At 21 mm, after the expanding front contacted the LCA, both branching points and coverage increased significantly again compared with previous stages. These data show that the coronary plexus undergoes consistent growth from 15 to 20 mm of body length and then enters a second expansion phase at 21 mm.

From 21 to 27 mm, the expanding ventral coronary network became increasingly dense and complex (Fig. 1R-U, Figs S2E-H and S3A-D). In contrast to previous stages, an increasing number of EGFP[+]/RFP[high] vessels were observed, suggesting progressive arterialization (Fig. S2E-H). At this stage, part of the ventral ventricular base surface remained consistently nonvascularized until fish reached 27 mm of body length (Fig. 1R-V). We define the developmental window spanning from 15 to 27 mm of body length as the expansion stage (Fig. S1A) and subdivide it into 'expansion I' (15-20 mm, first wave of coronary branching and coverage increase) and 'expansion II' (21-27 mm, second expansion wave and presence of the ventral ventricular base nonvascularized area).

From 28 to 30 mm, the ventral ventricular base nonvascularized area disappeared as the coronary network fully covered the ventricular surface (Fig. S2I-K) and no further increase in network coverage was observed (Fig. S3E-H). The ventricle presents a well-defined dorsal network of two to five large EGFP[+]/RFP[low] vessels, potentially main coronary veins, that in most instances expanded to reach the ventral surface (Fig. 1W-Y). Coronary arteries were well established with branches clearly visible on both ventricular surfaces (Fig. S2I-K). We define this stage as the maturation stage (Fig. S1A).

To determine the onset of coronary circulation, we performed O-dianisidine staining at different stages. Vessel-like structures were not clearly detectable until animals reached 18 mm of body length (Fig. 1Z, Fig. S4A,B). To further support this observation, we analyzed *TgBAC(etv2:EGFP)* ventricular sections and found that coronary vessels at this stage were indeed lumenized and contained DAPI[+] cells (Fig. 1AA). Next, we performed 5-ethynyl-2-deoxyuridine (EdU) incorporation assays on *TgBAC(etv2:EGFP)* ventricles before (16 mm) and after (19 mm) the onset of coronary perfusion. We found that the number of EdU[+] coronary ECs was increased at 19 mm compared with 16 mm (Fig. S4D-F), indicating that the onset of blood flow coincides with an increase in coronary EC proliferation.

### *TgBAC(sele:EGFP)* labels coronary veins

Several transgenic lines have been developed to study arterio-venous development in zebrafish. However, while arterial reporters remain active in the adult heart, venous marker expression becomes restricted to cardiac lymphatics, preventing visualization of coronary veins (Harrison et al., 2015; Marín-Juez et al., 2016, 2019). To circumvent this limitation, we generated a *TgBAC(sele:EGFP)* zebrafish line. *sele* encodes E-selectin, a cell adhesion molecule expressed on the EC surface (Collins et al., 1991). In zebrafish embryos, *sele* is highly expressed in the posterior cardinal vein (PCV) (Sun et al., 2015).

To determine whether this transgene is specifically expressed in venous ECs, we examined its expression throughout development in combination with *Tg(−0.8flt1:RFP)* and *Tg(dll4:TagRFP)*, both well-established arterial markers (Bussmann et al., 2010, 2011; Chong et al., 2011; Marín-Juez et al., 2016). We found strong *sele*:EGFP expression in venous vessels in adult fin rays (Fig. S5A) as well as in the PCV and venous intersegmental vessels (vISVs) at 5 dpf (Fig. S5B,C). We also analyzed *TgBAC(sele:EGFP)* in combination with *Tg(lyve1b:dsRed)* to visualize lymphatic vessels and found minimal EGFP expression in the thoracic duct (TD) at

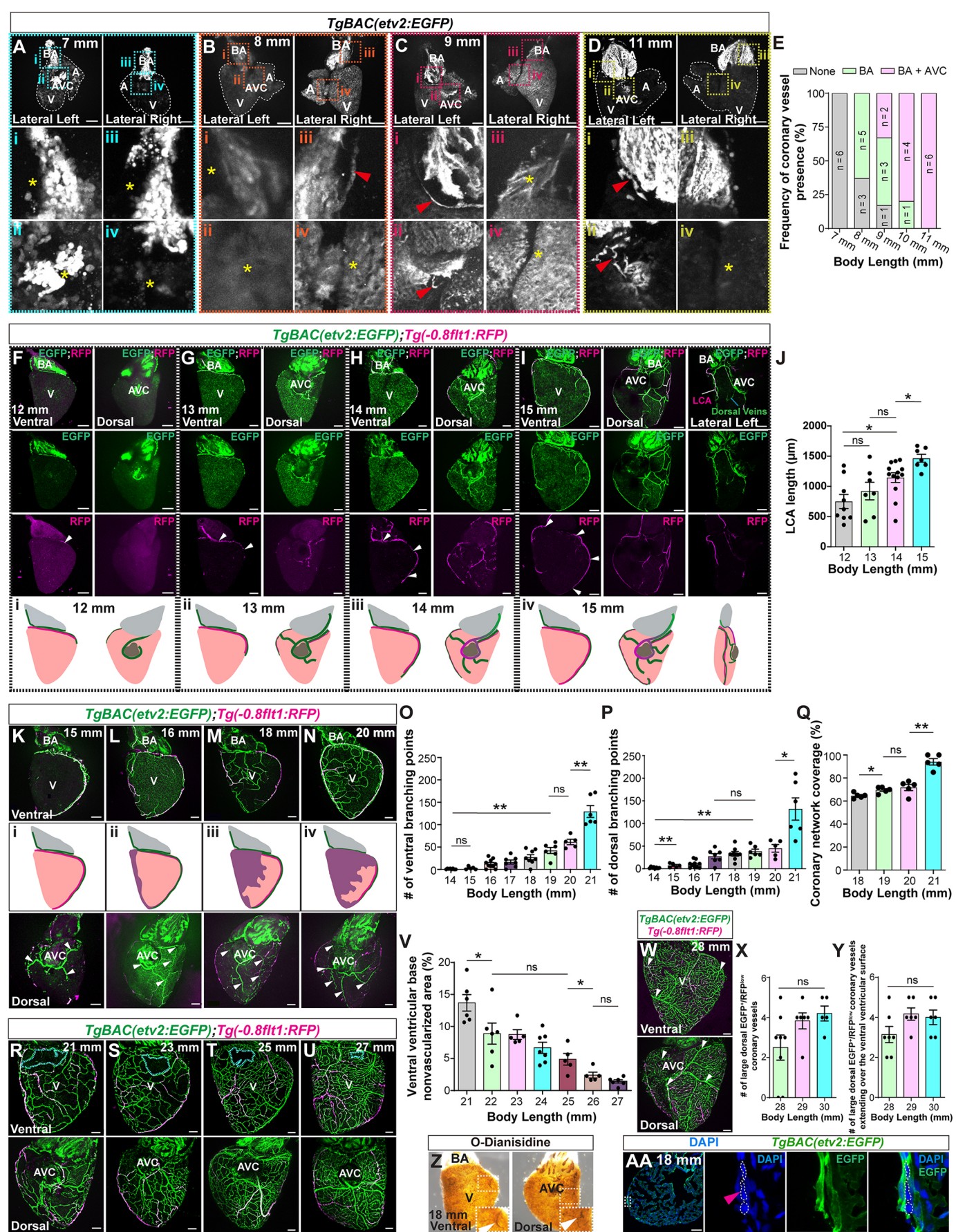

**Fig. 1.** See next page for legend.

**Fig. 1. Characterization of coronary development.** (A-D) Whole-mount *TgBAC(etv2:EGFP)* hearts from 7-mm (A), 8-mm (B), 9-mm (C) and 11-mm (D) -long zebrafish. High-magnification images of BA (i and iii) and AVC (ii and iv) regions are shown. Yellow asterisks indicate the absence of coronary vessels. Red arrowheads point to coronary vessels at the BA and AVC regions. (E) Frequency of coronary vessel presence in the BA and AVC regions in 7- to 11-mm-long zebrafish. (F-I) Whole-mount *TgBAC(etv2:EGFP)*; *Tg(−0.8flt1:RFP)* ventricles from 12-mm (F), 13-mm (G), 14-mm (H) and 15-mm (I) -long zebrafish. Schematic representations of coronary vessel development at each of the stages are shown below (i-iv). White arrowheads point to the LCA expanding across the left ventricular curvature from base to apex. (J) LCA length in 12- to 15-mm-long zebrafish. (K-N) Whole-mount *TgBAC(etv2:EGFP)*; *Tg(−0.8flt1:RFP)* ventricles from 15-mm (K), 16-mm (L), 18-mm (M) and 20-mm (N) -long zebrafish. Schematic representations of coronary network expansion (purple shadowed areas) are shown (i-iv). White arrowheads point to putative main dorsal veins. (O,P) Number of coronary branching points at the ventral (O) and dorsal (P) ventricular surfaces in 14- to 21-mm-long zebrafish. (Q) Percentage of coronary network coverage area at the ventral ventricular surface in 18- to 21-mm-long zebrafish. (R-U) Whole-mount *TgBAC(etv2:EGFP)*; *Tg(−0.8flt1:RFP)* ventricles from 21-mm (R), 23-mm (S), 25-mm (T) and 27-mm (U) -long zebrafish. Cyan dotted lines outline the nonvascularized area at the ventral ventricular base. (V) Nonvascularized area percentage at the ventral ventricular base in 21- to 27-mm-long zebrafish. (W) Whole-mount *TgBAC(etv2:EGFP)*; *Tg(−0.8flt1:RFP)* ventricle from a 28-mm-long zebrafish. White arrowheads point to EGFP$^+$/RFP$^{low}$ vessels. (X,Y) Number of large EGFP$^+$/RFP$^{low}$ coronary vessels on the dorsal ventricular surface (X) and extending over the ventral ventricular surface (Y). (Z) Erythrocytes in a whole-mount ventricle from an 18-mm-long zebrafish stained with O-dianisidine. Insets show magnifications of the boxed areas. White arrowheads point to vessel-like structures. (AA) *TgBAC(etv2:EGFP)* ventricular section from an 18-mm-long zebrafish stained for EGFP (coronary ECs, green) and DNA (DAPI, blue). Boxed area in left-hand image is shown at higher magnification on the right. Magenta arrowhead points to DAPI$^+$ cells in the lumen of an *etv2*: EGFP$^+$ coronary vessel. Dashed lines outline DAPI$^+$ cells in the lumen of an *etv2*:EGFP$^+$ coronary vessel. Data in graphs expressed as mean±s.e.m. ns, no significant difference. *$P<0.05$, **$P<0.01$ (two-tailed, Mann–Whitney *U* test). A, atrium; AVC, atrioventricular canal; BA, bulbus arteriosus; V, ventricle. Scale bars: 100 μm.

5 dpf (Fig. S5D). Next, we analyzed coronary vein and artery development from initiation to maturation stages in *TgBAC(sele: EGFP)*; *Tg(−0.8flt1:RFP)* ventricles. No vessel-like structures were found at 7 mm (Fig. 2A). At 8-11 mm (initiation stage), the first BA and AVC *sele*:EGFP$^+$ vessels were observed (Fig. 2B-C′), suggesting venous identity of these emerging vessels. At 13 mm of body length (establishment stage), two adjacent vessels, one EGFP$^+$/RFP$^−$ and another one EGFP$^−$/RFP$^{high}$ (the developing LCA), developed across the ventricular base and expanded toward the left ventricular curvature (Fig. 2D,D′). At this stage, the dorsal coronary plexus was composed of EGFP$^+$/RFP$^−$ and EGFP$^−$/ RFP$^{high}$ vessels. At 18 mm (expansion I stage), the venous plexus further developed dorsally and expanded ventrally following the previously observed right-to-left pattern (Fig. 2E). At this stage, some vessels near the expanding vascular front co-expressed both *sele*:EGFP and −*0.8flt1*:RFP, suggesting venous-to-capillary or arterial conversion (Hen et al., 2015; Red-Horse and Siekmann, 2019; Red-Horse et al., 2010; Xu et al., 2014) (Fig. 2E). At 24-28 mm (expansion II to maturation stages), the main coronary veins and arteries were distinguishable by marker expression, with *sele*: EGFP expression being restricted to coronary veins (Fig. 2F-G′). At this stage, the coronary veins primarily populated the dorsal ventricular half, with most extending ventrally, consistent with our data (Fig. 1W, Fig. S2I-K) and previous reports (Harrison et al., 2015; Marín-Juez et al., 2016).

Although no vessel-like structures were observed on the ventricle at 7 mm (Fig. 2A), we observed *sele*:EGFP$^+$/−*0.8flt1*:RFP$^−$ cells

on the BA. Previous studies have reported a population of BA lymphatic ECs at early developmental stages, before lymphatics populate the ventricle (Gancz et al., 2019; Harrison et al., 2019). Moreover, *sele*:EGFP was also lowly expressed in some TD *lyve1b*: dsRed$^+$ lymphatics (Fig. S5D). We analyzed *TgBAC(sele:EGFP)*; *Tg(lyve1b:dsRed)* ventricles and found that these cells co-express both reporters, indicating lymphatic identity (Fig. S5E,F). These data show that two distinct vascular populations co-exist on the BA at the initiation stage. Our analyses suggest that this population of BA lymphatics continue to expand at later stages (Fig. S5E-H), potentially reaching the ventricular surface during late expansion and maturation stages (Fig. S5I,J). Ventricular lymphatics only retained *lyve1b*:dsRed expression and closely followed coronary veins (Fig. S5J,K).

Collectively, these results show that *TgBAC(sele:EGFP)* is expressed by coronary venous ECs in zebrafish. This new marker enables dissection of venous identity, emergence and spatial distribution during coronary network development.

## Single-cell profiling identifies dynamic cellular transitions and markers defining distinct coronary subtypes

To better understand the cellular composition and interactions during heart development, we performed scRNA-seq on zebrafish ventricles at the initiation (10 mm), establishment (14 mm), expansion (18 mm) and maturation (28 mm) stages (Fig. S6A).

We identified and annotated all major cardiac cell types present at each developmental stage based on the expression of established marker genes (Fig. 3A,B, Fig. S6D). Our analysis revealed stage-specific changes in cellular composition across the four stages (Fig. 3C, Fig. S6B-F). Notably, coronary EC populations were lower at the initiation and establishment stages but showed a significant increase during the expansion and maturation stages. Similarly, macrophage and CM populations also increased progressively (Fig. 3C, Fig. S6E,F). We also observed changes in other cell populations, including EPDCs and mesenchymal cells (Fig. 3C, Fig. S6E,F). Interestingly, we identified two distinct populations of mural cells designated as 'mural cells 1' and 'mural cells 2'. The mural cells 1 number fluctuated throughout development, whereas mural cells 2 displayed a pattern similar to that of coronary ECs (Fig. 3C, Fig. S6E,F). To further resolve heterogeneity among cell populations potentially involved in coronary development, we performed secondary subclustering analyses of ECs, EPDCs and mural cells 2. EC subclustering identified five distinct subpopulations increasing from initiation to expansion stages, possibly reflecting progressive coronary vessel growth and remodeling (Fig. S7A,B). We also subclustered EPDCs and mural cells 2 and identified six EPDC subtypes and two mural cell 2 subtypes (Fig. S7C-F).

Marker gene expression analyses of the coronary EC subclustering identified arterial (*unc5b* and *efnb2*) (Adams et al., 1999; Gale et al., 2001; Navankasattusas et al., 2008), capillary (*fabp4a*) (van der Ark-Vonk et al., 2024), venous (*ephb4* and *sele*) (Adams et al., 1999; Sun et al., 2015; Wang et al., 1998) and lymphatic (*lyve1b* and *flt4*) (Jackson, 2004; Jerafi-Vider et al., 2021) ECs (Fig. 3D,E). These cell types increased from the initiation to the maturation stages (Fig. 3F) and displayed distinguishable gene expression signatures (Fig. 3G).

In addition, these analyses identified *tppp3*, *hpn*, *ano7*, *gig2j* and *gjb10* as new markers showing high and restricted expression in coronary ECs (Fig. 3H). Specifically, *tppp3*, *hpn* and *ano7* were enriched in capillaries, while *gig2j* and *gjb10* were mainly expressed in coronary veins (Fig. 3I). Using *in situ* hybridization chain reaction (HCR) coupled with immunostaining on whole-mount ventricles, we observed that *tppp3* and *ano7* were expressed mainly in capillary-like

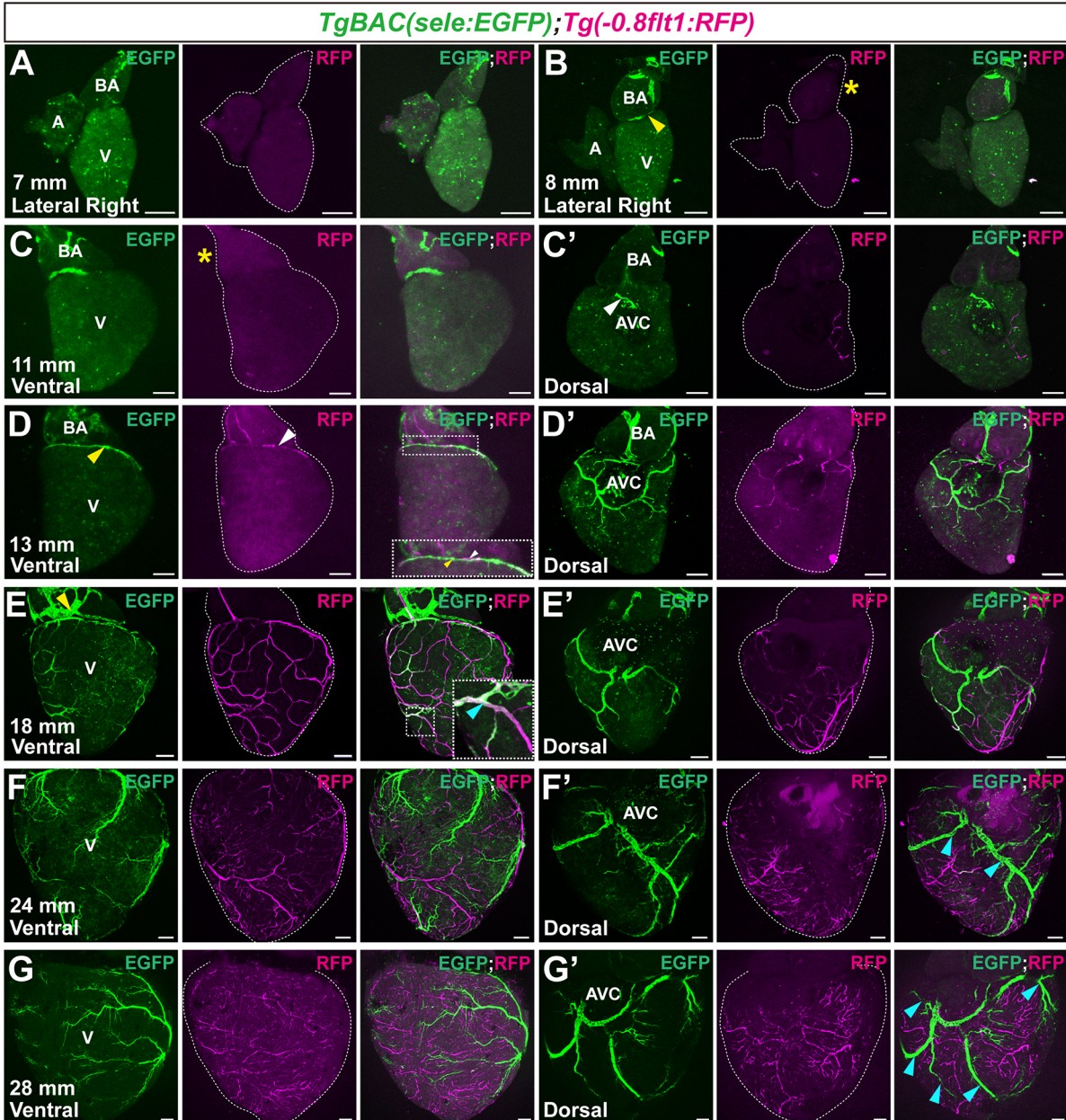

**Fig. 2. *TgBAC(sele:EGFP)* labels coronary veins during development.** (A) Whole-mount *TgBAC(sele:EGFP)*; *Tg(−0.8flt1:RFP)* heart from a 7-mm-long zebrafish. (B-C′) Whole-mount *TgBAC(sele:EGFP)*; *Tg(−0.8flt1:RFP)* hearts from 8-mm (B) and 11-mm (C,C′) -long zebrafish. Yellow arrowhead points to a *sele*:EGFP⁺ vessel in the BA-ventricular base junction. Yellow asterisks indicate absence of *−0.8flt1*:RFP expression. White arrowhead points to a *sele*: EGFP⁺ vessel around the AVC. (D-E′) Whole-mount *TgBAC(sele:EGFP)*; *Tg(−0.8flt1:RFP)* ventricles from 13-mm (D,D′) and 18-mm (E,E′) -long zebrafish. (D) Yellow arrowhead points to an EGFP⁺/RFP⁻ (vein) vessel. White arrowhead points to an EGFP⁻/RFPhigh (LCA, artery) vessel. (E) Yellow arrowhead points to BA EGFP⁺ lymphatics. Cyan arrowhead points to an EGFP⁺/RFP⁺ vessel. (F-G′) Whole-mount *TgBAC(sele:EGFP)*; *Tg(−0.8flt1:RFP)* ventricles from 24-mm (F,F′) and 28-mm (G,G′) -long zebrafish. Cyan arrowheads point to main dorsal coronary veins. Insets show magnifications of the boxed areas. A, atrium; AVC, atrioventricular canal; BA, bulbus arteriosus; V, ventricle. Scale bars: 100 μm.

vessels and some large veins (*etv2*:EGFP⁺/−*0.8flt1*:RFP⁻) (Fig. 3J,K). *gig2j* and *gjb10* were expressed in main dorsal veins (Fig. 3L-O). *hpn* levels were not clearly detectable by HCR.

## Manipulation of Vegfa signaling impairs coronary network formation

Vegfa is a key regulator of angiogenesis and its depletion leads to severe vascular and developmental defects as well as embryonic lethality across vertebrate species (Carmeliet et al., 1999; Hudlicka et al., 1992; Rossi et al., 2016; Taimeh et al., 2013), including

zebrafish (Rossi et al., 2016). We have previously shown that *vegfaa* mutants can be rescued to adulthood, providing a unique model in which to study the role of Vegfaa in coronary development (Marín-Juez et al., 2016). At 14 mm (establishment stage), rescued *vegfaa⁻/⁻* exhibited defective LCA and main dorsal vein growth compared to wild-type (WT) fish (Fig. 4A-C). At 18 mm (expansion I), dorsal and ventral vessel expansion was severely reduced in rescued *vegfaa⁻/⁻* (Fig. 4D-H). In addition, we noticed that LCA localization was altered in several *vegfaa⁻/⁻* ventricles, with the LCA growing along the right ventricular curvature (Fig. 4F,I). At

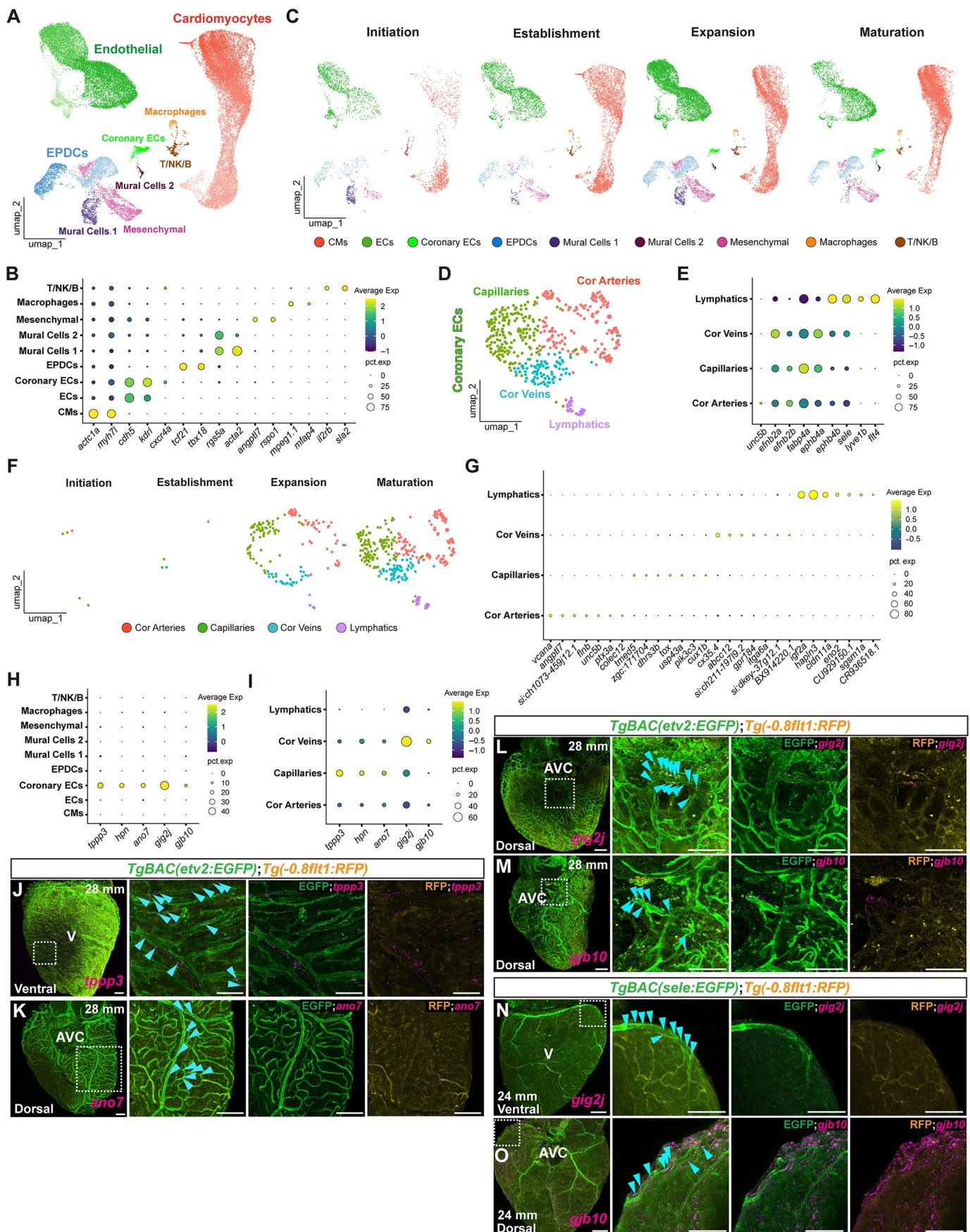

**Fig. 3.** See next page for legend.

**Fig. 3. Single-cell transcriptome atlas during coronary development.**
(A-C) Combined uniform manifold approximation and projection (UMAP) plot
(A) and split UMAP plots (C) visualizing nine cell types identified across
cardiac development and dot plot showing average marker gene expression
and abundance corresponding to each cell type (B). (D-F) Combined UMAP
plot (D) and split UMAP plots (F) of coronary ECs visualizing four subtypes
identified across cardiac development and dot plot showing average
expression and abundance of selected marker genes corresponding to each
subtype (E). (G) Dot plot showing average expression and abundance of top
seven marker genes in coronary ECs. (H,I) Dot plots showing average
expression and abundance of selected highly expressed marker genes of
coronary ECs in all cell types (H) and in coronary EC subtypes (I).
(J-M) In situ HCR for tppp3 (J), ano7 (K), gig2j (L), gjb10 (M) coupled with
immunostaining for EGFP (green) and RFP (yellow) on whole-mount
TgBAC(etv2:EGFP); Tg(−0.8flt1:RFP) ventricles. Cyan arrowheads point to
EGFP⁺/RFPˡᵒʷ or EGFP⁺/RFP⁻ showing HCR signal. AVC, atrioventricular
canal; V, ventricle. (N,O) In situ HCR for gig2j (N), gjb10 (O) coupled with
immunostaining for EGFP (green) and RFP (yellow) on whole-mount
TgBAC(sele:EGFP); Tg(−0.8flt1:RFP) ventricles. Cyan arrowheads point to
EGFP⁺/RFPˡᵒʷ or EGFP⁺/RFP⁻ showing HCR signal. Dotted boxes indicate
the regions shown at higher magnification to the right. T/NK/B, T cells/natural
killer cells/B cells. Scale bars: 100 µm.

24 mm (expansion II), the $vegfaa^{-/-}$ coronary network expanded and
reached coverage levels similar to those in WTs (Fig. 4J-M). While at
this stage the overall ventricular coverage was not significantly
different, rescued $vegfaa^{-/-}$ ventricles exhibited a greater degree of
variability (Fig. 4J,N). To further analyze these differences, we
evaluated the coronary network distribution in WT and mutant
ventricles. To assess their spatial distribution, we divided the ventral
ventricular surface into four quadrants and measured vascular
coverage in each of these areas. While WT fish displayed relatively
uniform coronary network distribution among the four quadrants,
rescued $vegfaa^{-/-}$ clustered coronaries in area I and displayed sparse
coverage in area IV (Fig. 4J,N). At 28 mm (maturation), rescued
$vegfaa^{-/-}$ exhibited incomplete and reduced coronary coverage,
characterized by thinner vessels and a disorganized vascular network
compared to WT siblings (Fig. 4O-R), consistent with previous
findings (Marín-Juez et al., 2016). Vegfa regulates endothelial
development and specification (Carmeliet et al., 1996; Jin et al.,
2017). To analyze possible arterial phenotypes, we analyzed $vegfaa^{-/-}$
$Tg(−0.8flt1:RFP)$ ventricles. $vegfaa^{-/-}$ ventricles from establishment
to maturation showed reduced and sparse RFP expression, suggesting
either defects in arterialization or dysregulation of transgene expression
(Fig. S8A-D).

Since we observed a severe reduction in LCA growth in $vegfaa^{-/-}$
ventricles, we hypothesized that defective proliferation might be
the underlying cause for this defect. To test this possibility, we
performed EdU incorporation assays coupled with immunostaining
in whole-mount $TgBAC(etv2:EGFP)$ ventricles from 15 mm fish.
We found a significant reduction in the percentage of EdU⁺ coronary
ECs within the LCA of $vegfaa^{-/-}$ compared to WT (Fig. 4S,T).
Moreover, our data indicate LCA mislocalization in some $vegfaa^{-/-}$
ventricles (Fig. 4F,I). This phenotype could be due to alterations
in developmental patterning caused by defective Vegfa signaling.
To test this possibility, we analyzed LCA development in $scube2^{-/-}$
fish. Scube2 modulates Vegfa signaling to induce angiogenesis
(Ali, 2020; Lin et al., 2017) and functions upstream of Sonic
Hedgehog (Shh) signaling during developmental patterning
(Kawakami et al., 2005; Woods and Talbot, 2005). At 13 mm,
$scube2^{-/-}$ exhibited defective LCA and dorsal coronary vein growth,
recapitulating $vegfaa^{-/-}$ phenotypes (Fig. 4U-X). Furthermore, 50%
of $scube2^{-/-}$ display mislocalized LCA positioning, like $vegfaa^{-/-}$
ventricles (Fig. 4W,Y). Together, these data indicate that Vegfa
signaling regulates LCA growth and localization.

As retinoic acid (RA) signaling regulates endothelial proliferation
via Vegf and Shh pathways (Bohnsack et al., 2004; Lai et al., 2003),
we next examined the expression of RA signaling-related genes in
our scRNA-seq datasets. We found that several RA pathway
components, including $aldh1a2$, $raraa$, $rarab$, $rxraa$ and $rxrab$ were
highly expressed in ECs, coronary ECs, EPDCs, mesenchymal
cells, mural cells and immune cell populations mainly during
expansion stage. The expression of genes encoding RA-degrading
enzymes ($cyp26a1$ and $cyp26c1$) was not detectable (Fig. S9).

Recent studies have shown that VEGF can either promote
or inhibit angiogenesis in a dose-dependent manner (Pontes-
Quero et al., 2019). Similarly, stimulation of Vegfa signaling
through $flt1$ deletion enhances heart regeneration while strong
Vegfaa overexpression impairs it (Karra et al., 2018; Wang et al.,
2024). To assess how different levels of Vegfa signaling induction
regulate coronary development, we analyzed $flt1$ mutants and the
$Tg(myl7:CreER)$; $Tg(βactin2:loxP-mTagBFP-STOP-loxP-vegfaa)$
overexpression line [hereafter referred to as $Tg(myl7:CreER)$;
$Tg(βact2:BS-vegfaa)$] at the different stages.

First, we analyzed $flt1^{-/-}$ at the expansion I stage (18 mm). At this
stage, there was no difference in coronary coverage between $flt1^{-/-}$
and WT siblings (Fig. 5A-C). However, the directional expansion of
the coronary network was altered. In $flt1^{-/-}$ fish, the coronary network
predominantly expanded left to right, contrasting with the right-to-left
pattern observed in WTs (Fig. 5A,D). At the expansion II stage
(24 mm), vessel coverage remained similar in $flt1^{-/-}$ and WT siblings
(Fig. 5E-G), yet the spatial distribution of the coronary network in
$flt1^{-/-}$ fish was irregular. Specifically, areas I and III of the ventral
ventricular surface appeared hypervascularized in $flt1^{-/-}$ (Fig. 5E,I),
resulting in a significant reduction of the ventral ventricular base
nonvascularized area (Fig. 5E,H). At the maturation stage (28 mm),
no differences in total coronary coverage were observed between
$flt1^{-/-}$ and WT siblings (Fig. S10A-D). $Tg(−0.8flt1:RFP)$ expression
was strongly upregulated in $flt1^{-/-}$ coronary ECs (Fig. S8E-G).

Next, to further test the effect of strong Vegfa signaling
stimulation, we overexpressed $vegfaa$ using the $Tg(myl7:CreER)$;
$Tg(βact2:BS-vegfaa)$ line crossed with the $TgBAC(etv2:EGFP)$;
$Tg(−0.8flt1:RFP)$ background. Following tamoxifen administration,
treated ventricles displayed pronounced hypervascularization
and distorted vascular architecture compared to control ventricles
(Fig. 5J-L). Moreover, $vegfaa$ overexpression led to a threefold
dilation of the main dorsal coronary veins (Fig. 5J,M). As no distinctive
LCA could be observed after $vegfaa$ overexpression, we analyzed
$Tg(−0.8flt1:RFP)$ ventricles. As expected, based on observations
in $flt1^{-/-}$ ventricles, RFP expression was strongly upregulated in
all coronary vessels, preventing us from assessing arterial identity
(Fig. S8H). Interestingly, $−0.8flt1$:RFP was ectopically expressed
in endocardial cells in recombined $Tg(myl7:CreER)$; $Tg(βact2:BS-
vegfaa)$ ventricles, likely reflecting the strong induction of Vegfaa in
these fish (Fig. S8H). These data further support that the changes in
$−0.8flt1$:RFP expression observed upon Vegfa manipulation result, at
least in part, from altered $vegfaa$ expression levels, rather than reflecting
changes in specification.

Lastly, we examined Vegf receptor expression across distinct
coronary EC subtypes in our scRNA-seq datasets. $flt1$ and $kdrl$ were
expressed in all coronary ECs with lower expression in lymphatic
ECs, $kdr$ was expressed in all clusters and $flt4$ expression was
largely restricted to lymphatic ECs (Fig. S11).

Collectively, our results show that Vegfa signaling does not simply
control coronary vessel growth, but also dictates specific aspects of
network patterning, including LCA positioning, directional expansion
across the ventricle, and regional distribution of vessel density.

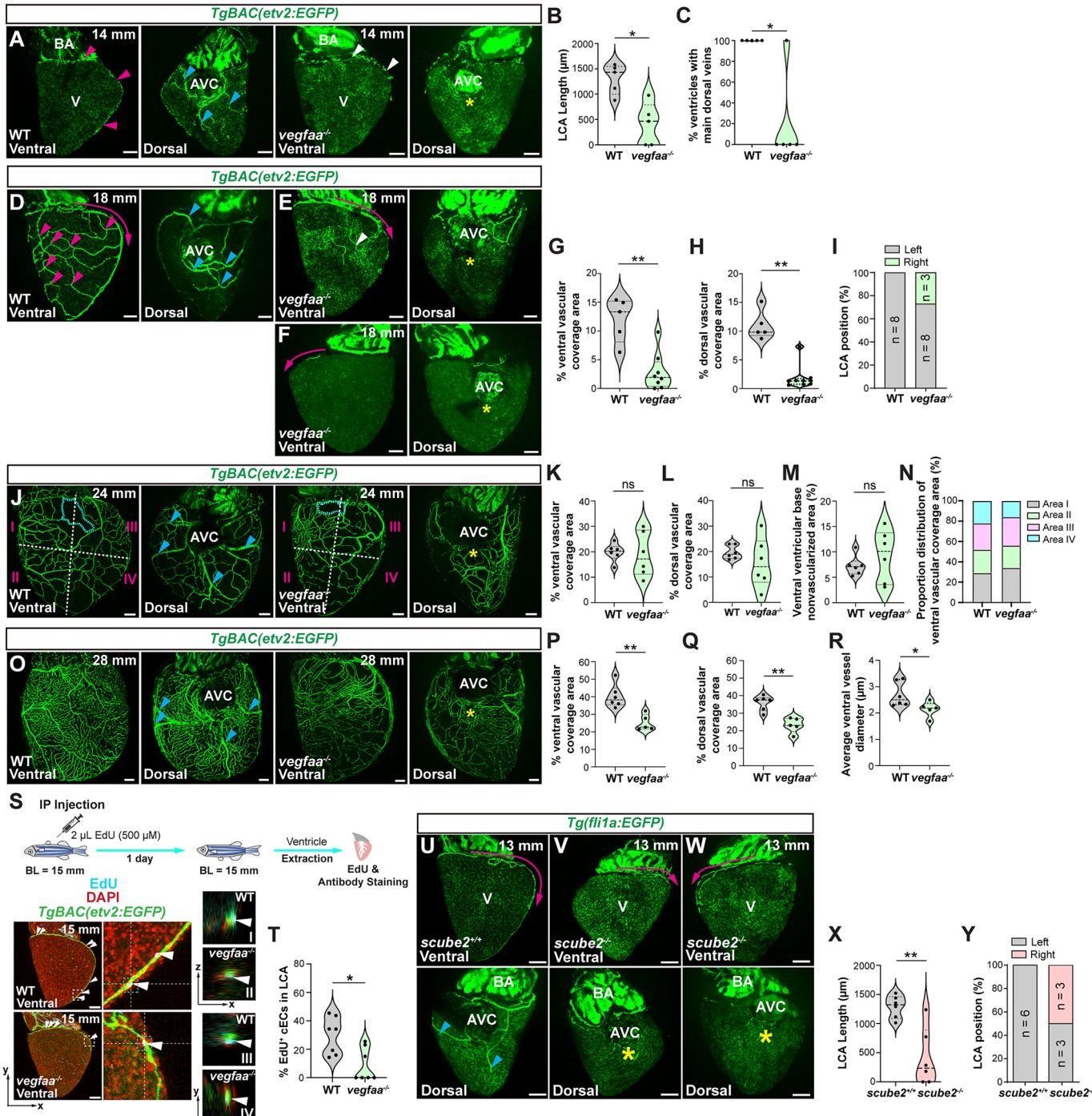

**Fig. 4.** See next page for legend.

## Spatiotemporal dynamics of coronary vessel–CM interactions during heart development

Previous studies have shown that coronary vessels act as scaffolds for growing CMs during development and regeneration in zebrafish (Marín-Juez et al., 2019). To profile this interaction, we used double-transgenic *Tg(−0.8flt1:RFP)*; *Tg(gata4:GFP)* fish to label coronary ECs and growing cortical CMs, respectively (Gupta and Poss, 2012; Kikuchi et al., 2010; Marín-Juez et al., 2019). At 14 mm, we observed that *gata4*:GFP⁺ CMs emerged along the ventricular base, the left ventricular curvature and near the AVC region (Fig. 6A), consistent with previous reports (Gupta et al., 2013). Exclusively at

this stage, we observed *gata4*:GFP⁺ cells with epicardial morphology sparsely distributed over the ventricle (Fig. 6A). We performed Caveolin-1 (Cav-1) staining, an epicardial marker, on *Tg(gata4: GFP)* ventricular sections and found that these cells were Cav-1⁺, indicating epicardial identity (Fig. S12A). At 18 mm, *gata4*:GFP⁺ cortical CMs populated the ventral surface, paralleling the coronary network expansion and showing an intimate association with the developing vasculature (Fig. 6B,B′). To assess the spatial relationship between coronary vessels and CM growth, we quantified ventral CM coverage relative to coronary vessel proximity, grouping distances into 0-15 μm, 15-30 μm and >30 μm. The majority of growing CMs

**Fig. 4. Loss of Vegfa signaling impairs coronary network formation.**
(A) Whole-mount *TgBAC(etv2:EGFP)* ventricles from 14-mm-long WT and
rescued *vegfaa*$^{-/-}$. Magenta arrowheads point to the LCA, blue arrowheads
point to main dorsal coronary veins, white arrowheads point to defective
LCA, and yellow asterisk indicates absence of main dorsal coronary veins.
(B,C) LCA length (B) and percentage of ventricles with main dorsal coronary
veins (C) in WT and rescued *vegfaa*$^{-/-}$ 14-mm-long zebrafish. (D-F) Whole-
mount *TgBAC(etv2:EGFP)* ventricles from 18-mm-long WT (D) and
*vegfaa*$^{-/-}$ (E,F). Magenta arrows indicate the direction and position of LCA
growth. Magenta arrowheads point to coronary plexus branching points.
Blue arrowheads point to main dorsal coronary veins. White arrowhead
points to defective ventral vasculature. Yellow asterisks indicate absence of
main dorsal coronary veins. (G-I) Percentage of the ventral (G) and dorsal
(H) vessel coverage, and LCA position (I). (J) Whole-mount *TgBAC(etv2:
EGFP)* ventricles from 24-mm-long WT and rescued *vegfaa*$^{-/-}$. Cyan dotted
lines outline the nonvascularized area at the ventral ventricular base. Blue
arrowheads point to main dorsal coronary veins. Yellow asterisk indicates
absence of large dorsal coronary veins. White dashed lines divide the
ventricular surface into four areas. (K-N) Percentage of the ventral (K) and
dorsal (L) vessel coverage, ventral ventricular base nonvascularized area
(M) and proportion distribution of ventral vascular coverage (N) in 24-mm-
long WT and rescued *vegfaa*$^{-/-}$. (O) Whole-mount *TgBAC(etv2:EGFP)*
ventricles from 28-mm-long WT and rescued *vegfaa*$^{-/-}$. Blue arrowheads
point to main dorsal coronary veins. Yellow asterisk indicates absence of
large dorsal coronary veins. (P-R) Percentage of the ventral (P) and dorsal
(Q) vessel coverage, and average ventral vessel diameter (R). (S) Whole-
mount *TgBAC(etv2:EGFP)* ventricles of WT and rescued *vegfaa*$^{-/-}$
ventricles from 15-mm-long zebrafish stained for EGFP (coronary ECs,
green), EdU (proliferating cells, cyan) and DNA (DAPI, red). White
arrowheads point to EdU$^+$/EGFP$^+$ cells in the LCA. *xz* axis orthogonal views
of WT (I) and rescued *vegfaa*$^{-/-}$ (II) ventricles and *yz* axis orthogonal views
of WT (III) and rescued *vegfaa*$^{-/-}$ (IV) ventricles are shown. Schematic
summarizes the experimental protocol. BL, body length; IP, intraperitoneal.
(T) Percentage of EdU$^+$/EGFP$^+$ cells in the LCA in 15-mm-long WT and
rescued *vegfaa*$^{-/-}$. (U-W) Whole-mount *Tg(fli1a:EGFP)* ventricles from
*scube2*$^{+/+}$ (U) and *scube2*$^{-/-}$ (V,W) 13-mm-long zebrafish. Magenta arrows
indicate the direction and position of LCA growth. Blue arrowheads point to
main dorsal coronary veins, and yellow asterisks indicate absence of main
dorsal coronary veins. (X,Y) LCA length (X) and percentage position (Y) in
13-mm-long *scube2*$^{+/+}$ and *scube2*$^{-/-}$. Data in graphs are expressed as
mean±s.e.m. ns, no significant difference. *$P<0.05$, **$P<0.01$ (two-tailed,
unpaired Student's *t*-test). AVC, atrioventricular canal; BA, bulbus arteriosus;
V, ventricle. Scale bars: 100 µm.

were located within 0-15 µm of developing coronaries (Fig. 6C),
suggesting close spatial coordination.

Next, we performed whole-mount cardiac troponin T (cTnT)
staining in *Tg(−0.8flt1:RFP)*; *Tg(gata4:GFP)* ventricles to analyze
sarcomere arrangement. CMs trailing behind the expanding front
displayed defined sarcomeres arranged perpendicularly to near
coronaries, indicating CM orientation and structural maturation. In
contrast, CMs at the expanding front exhibited poorly defined
sarcomeres (Fig. 6D). To test whether the spatial proximity between
CMs and coronary ECs reflects coordinated growth dynamics, we
performed EdU incorporation assays combined with immunostaining.
We observed EdU$^+$ cells in both expanding coronary vessels and
adjacent CMs, indicating synchronized proliferation (Fig. 6Ei). EdU$^+$
CMs were consistently localized near developing coronary vessels
(Fig. 6Eii). Moreover, EdU$^+$ CMs were enriched at the expanding
front (Fig. 6Eiii), whereas most EdU$^+$ cells behind were *gata4*:GFP$^-$
(Fig. 6Eiv). These findings suggest that coronary vessels and
neighboring CMs undergo coordinated proliferation specifically at
the myo-vascular expanding front.

At the expansion II stage (25 mm), *gata4*:GFP$^+$ CMs continued
to expand over the ventricular surface (Fig. 6F). Notably, the ventral
ventricular base, the last area to be vascularized (Fig. 1R-U), was
also the last area to be populated by *gata4*:GFP$^+$ CMs, further
indicating coronary-CM coordination (Fig. 6F). At the maturation

stage (28 mm), the entire ventricular surface was populated by both
cortical CMs and coronary vessels (Fig. 6G). Overall, these data
show that the developing coronary network functions as a vascular
scaffold for cortical CMs, with the spatial expansion of these CMs
mirroring that of developing coronaries (Fig. S13A).

To better understand this interaction, we first performed secondary
CM subclustering of our scRNA-seq datasets and identified seven
CM subtypes. Primordial CMs (pCMs) were the least abundant and
characterized by high expression of *acta2*, *hey2*, *actn1* and *smoc1*
(Carey et al., 2024; Tsedeke et al., 2021) (Fig. 7A,B). Trabecular
CMs (tCMs) with three subclusters (tCMs-1, tCMs-2 and tCMs-3)
were the most abundant subtype and displayed strong expression of
the canonical myocardial markers *myl7* and *actc1a*, which are
associated with contractile differentiated CMs (Carey et al., 2024)
(Fig. 7A,B). Cortical CMs (cCMs) with two subclusters (cCMs-1 and
cCMs-2) expressed lower levels of *myl7* and *actc1a*, and lacked
expression of *tbx5a*, as previously described (Carey et al., 2024;
Sánchez-Iranzo et al., 2018) (Fig. 7A,B). We also identified a
CM population that co-expressed markers of both primordial
(*actn1*, *smoc1*) and trabecular (*tbx5a*, *gata6*, *mef2d*) CMs. Given
this hybrid expression profile, we designated this cluster as
primordial-trabecular CMs (p-tCMs), suggesting a transitional or
intermediate CM state (Fig. 7A,B). Across developmental stages, we
observed dynamic shifts in CM subtype abundance. Both pCMs and
p-tCMs increased from initiation to expansion stages then declined at
maturation. tCMs followed a similar trend. In contrast, cCMs
expanded progressively from initiation to maturation (Fig. 7C),
consistent with cortical wall formation (Gupta and Poss, 2012).

Next, we performed cell–cell communication analyses between
coronary ECs and the seven CM clusters (Figs S6D and S14A,B).
Coronary ECs appeared to mainly interact with pCMs and p-tCMs.
The interaction strength between these cell types increased gradually
from initiation to establishment then decreased at later stages (Fig. 7D).
We also found interactions among pCMs, p-tCMs and cCMs-1
throughout development (Fig. 7D). Other cardiac components,
including EPDCs, also appeared to interact with coronary ECs and
CMs. Likewise, macrophage interactions with coronary ECs and CMs
were observed throughout development (Fig. S14C).

As we found that coronary ECs mainly interact with pCMs and
p-tCMs, we performed ligand–receptor analysis between these cell
types. Within coronary ECs, we observed higher number of ligand–
receptor pairs at expansion and maturation stages (Fig. 7G,H).
Coronary EC-pCM ligand–receptor pairs decreased gradually from
initiation to maturation stages, while coronary EC-p-tCM ligand–
receptor pairs increased from initiation to establishment stages
(Fig. 7E-H). Notably, the *hbegfa–erbb4b* interaction was predicted
to be strong between coronary ECs, and pCMs and p-tCMs, from
initiation to establishment stages (Fig. 7E-H). EC-secreted HB-EGF
promotes CM proliferation, survival and maturation (Iwamoto and
Mekada, 2006; Iwamoto et al., 2003). In addition, the *nectin3b–pvrl2l*
heterophilic interaction was predicted exclusively during initiation
stage, between coronary ECs and pCMs (Fig. 7E), whereas the
*pvrl2l–pvrl2l* homophilic interaction observed in coronary ECs and
coronary ECs-pCMs peaked at the establishment stage (Fig. 7F-H).
Pvrl2l (nectin-2-like) mediates Ca$^{2+}$-independent cell–cell adhesion,
suggesting a role in supporting physical integration and alignment
of coronary ECs and CMs during vessel expansion (Kinugasa
et al., 2012; Takai et al., 2003). To assess whether subtype-specific
interactions correlate with proliferation, we examined cell cycle-
related gene expression. Proliferation markers including *mki67*, *pcna*,
*aurka* and *mcm5* were enriched from initiation to expansion stages,
particularly in pCMs and p-tCMs (Fig. S15).

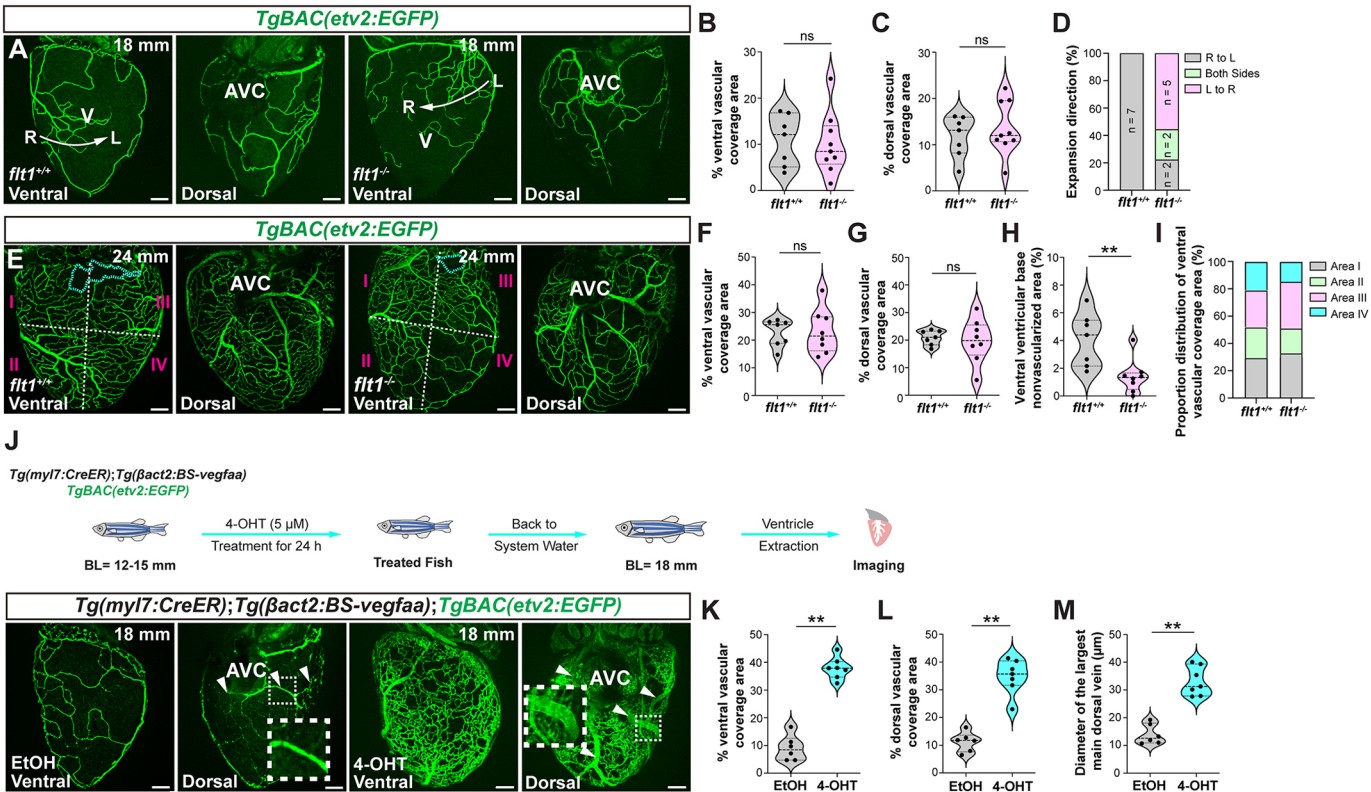

**Fig. 5. Coronary network development in conditions of increased Vegfaa signaling.** (A) Whole-mount *TgBAC(etv2:EGFP)* ventricles from 18-mm-long *flt1*[+/+] and *flt1*[−/−]. White arrows from R (right) and L (left) represent the direction of coronary network expansion. (B-D) Percentage of the ventral (B) and dorsal (C) vessel coverage, and direction of coronary network expansion (D) in 18-mm-long *flt1*[+/+] and *flt1*[−/−]. (E) Whole-mount *TgBAC(etv2:EGFP)* ventricles from 24-mm-long *flt1*[+/+] and *flt1*[−/−]. Cyan dotted lines outline the nonvascularized area at the ventral ventricular base. White dashed lines divide the ventricular surface into four areas. (F-I) Percentage of the ventral (F) and dorsal (G) vessel coverage, ventral ventricular base nonvascularized area (H) and proportion distribution of ventral vascular coverage (I) in 24-mm-long *flt1*[+/+] and *flt1*[−/−]. (J) Whole-mount *Tg(myl7:CreER)*; *Tg(βact2:BS-vegfaa)*; *TgBAC(etv2: EGFP)* ventricles from 18-mm-long control (EtOH, ethanol) and tamoxifen (4-OHT) treated (*vegfaa*[OE]) fish. Dotted boxes indicate the regions shown at higher magnification in insets. White arrowheads point to main dorsal coronary veins. Schematic summarizes the experimental protocol. BL, body length. (K-M) Percentage of the ventral (K) and dorsal (L) vessel coverage, and diameter of the largest main dorsal coronary vein (M) in 18-mm-long control and 4-OHT-treated zebrafish. Data in graphs are expressed as mean±s.e.m. ns, no significant difference. **P<0.01 (two-tailed, unpaired Student's *t*-test). AVC, atrioventricular canal; V, ventricle. Scale bars: 100 μm.

Altogether, these data identify complex signaling interactions between coronary ECs and CMs during cardiac development.

## Impaired coronary vessel formation via Vegfaa manipulation disrupts CM expansion

Next, we used our genetic models for Vegfa signaling manipulation to test how alterations in coronary development affect CM growth. We focused on the expansion stage, characterized by strong coronary-cortical CM interplay and active growth. At this stage, *vegfaa*[−/−] fish did not follow the typical right-to-left CM expansion, displaying instead a more symmetric growth pattern (Fig. 8A,B). CM expansion was also altered in both *flt1*[−/−] and *Tg(myl7:CreER)*; *Tg(βact2:BS-vegfaa)* tamoxifen-treated (*vegfaa*[OE]) ventricles (Fig. 8C,D). While ventral CM coverage was unchanged between WT and *vegfaa*[−/−] fish, it appeared to be increased in *flt1*[−/−] ventricles (Fig. 8A-C,E), consistent with previous studies showing enhanced CM regeneration in these mutants (Wang et al., 2024). Interestingly, inducing higher *vegfaa* levels using the *Tg(myl7:CreER)*; *Tg(βact2:BS-vegfaa)* line led to a significant reduction in total CM coverage (Fig. 8D,E). Next, we analyzed differences in CM distribution. In WT fish, CM coverage was predominant in areas I and II, mirroring the coronary expansion pattern (Fig. 8A,F). In contrast, *vegfaa*[−/−] and *flt1*[−/−] fish displayed reduced CM coverage in these areas with increased coverage of areas III and IV. *vegfaa* overexpression using

the *Tg(myl7:CreER)*; *Tg(βact2:BS-vegfaa)* line shifted CM expansion distribution toward areas I and III, indicating alterations in regional expansion (Fig. 8D,F). It is possible that *vegfaa* overexpression in *Tg(myl7:CreER)*; *Tg(βact2:BS-vegfaa)* fish impairs the mitogenic potential of Vegfaa, leading to mislocalized and restricted CM expansion as reported during regeneration (Karra et al., 2018). These data suggest that precise coronary developmental patterning, orchestrated by Vegfa signaling, is essential for regulating CM growth and expansion to populate the ventricular surface.

To further determine whether CM proliferation was affected under conditions of impaired coronary development, we performed EdU incorporation assays coupled with immunostaining in *Tg(gata4: GFP)* *vegfaa*[−/−] mutants. In WT fish, EdU[+] CMs were significantly enriched on the right side of the ventral ventricular surface. This spatial asymmetry was lost in *vegfaa*[−/−] fish accompanied by a significant reduction in the total number of EdU[+] CMs (Fig. 8G,H).

Overall, these findings indicate that coronary vessel development is coupled with CM expansion, and that disruption of coronary growth through impaired Vegfa signaling compromises cortical myocardial expansion.

## DISCUSSION

We assemble a developmental roadmap for coronary vasculature formation and show how coronary vessels play an instructive role

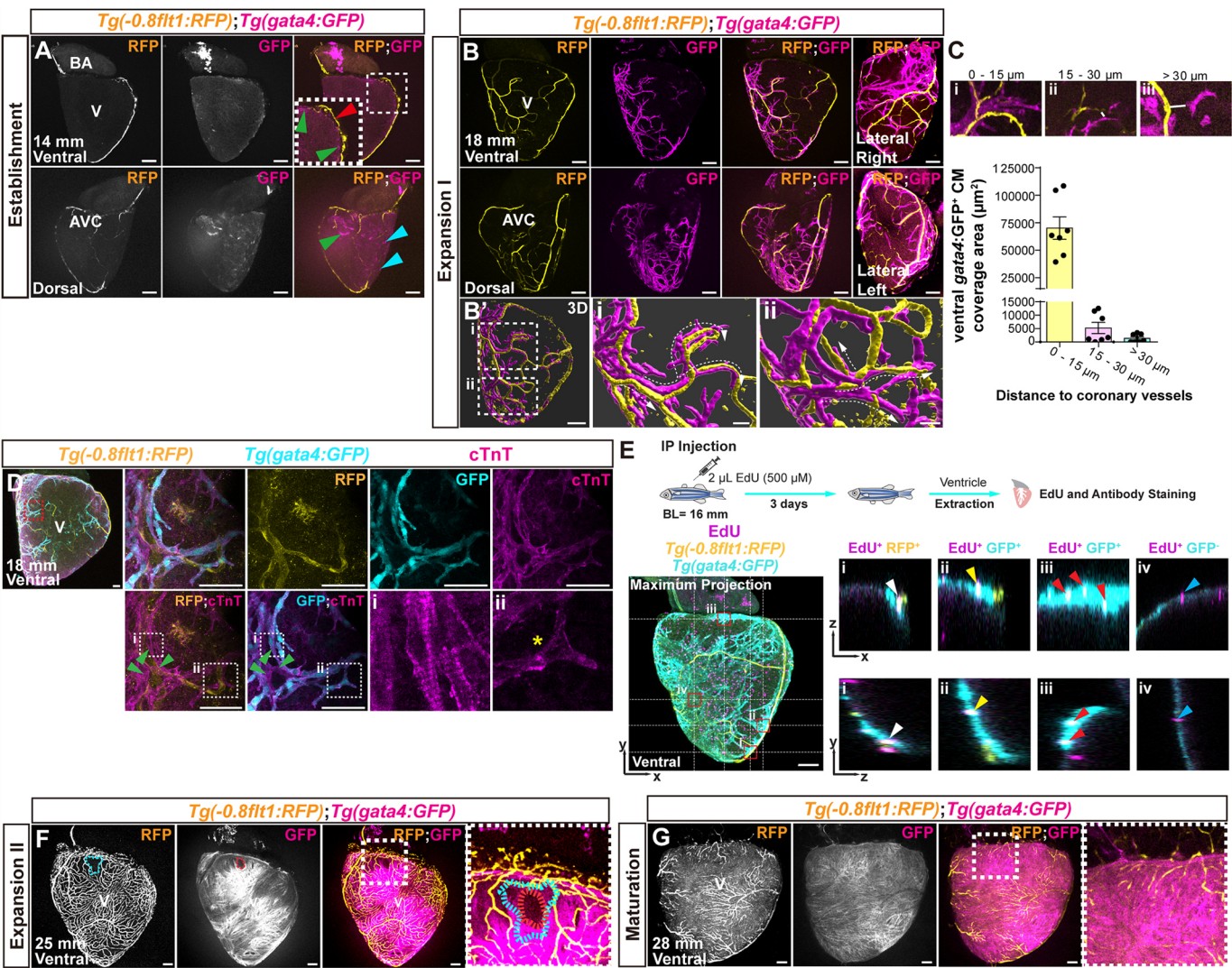

**Fig. 6. Coronary vessel–CM interactions during development.** (A) Whole-mount *Tg(−0.8flt1:RFP)*; *Tg(gata4:GFP)* ventricle from a 14-mm-long zebrafish. Red arrowhead points to the LCA. Green arrowheads point to GFP+ cortical CMs. Inset shows magnified view of the boxed region. Cyan arrowheads point to morphologically distinct GFP+ cells. (B,B′) Whole-mount a *Tg(−0.8flt1:RFP)*; *Tg(gata4:GFP)* ventricle from an 18-mm-long zebrafish showing expanding GFP+ cortical CMs in close proximity to developing coronary vessels (B) and a 3D reconstruction of the ventral ventricular surface (B′). High-magnification images of the ventral 3D reconstruction are shown (i,ii). White dashed arrows indicate CM growth expansion toward the vascular front. (C) Ventral ventricular cortical coverage area of *gata4*:GFP+ CMs located at 0-15, 15-30 and >30 µm from the nearest RFP+ coronary vessel. Example images are shown for each of the categories with the relevant distance marked by white bars (i-iii). (D) Whole-mount *Tg(−0.8flt1:RFP)*; *Tg(gata4:GFP)* ventricle from an 18-mm-long zebrafish stained for RFP (coronary ECs, yellow), GFP (growing cortical CMs, cyan) and cardiac troponin T (cTnT, sarcomeres, magenta). Green arrowheads point to *gata4*:GFPlow CMs displaying well-organized sarcomeres. Yellow asterisk marks *gata4*:GFP+ CMs at the expanding vascular front lacking defined sarcomeres. (E) Whole-mount *Tg(−0.8flt1:RFP)*; *Tg(gata4:GFP)* zebrafish ventricle stained for RFP (coronary ECs, yellow), GFP (growing cortical CMs, cyan) and EdU (proliferating cells, magenta). *xz* and *yz* orthogonal views show proliferating coronary ECs (i; EdU+/RFP+, white arrowheads), proliferating CMs in close proximity to coronaries (ii; EdU+/GFP+, yellow arrowheads), proliferating CMs near an expanding vascular front (iii; EdU+/GFP+, red arrowheads) and GFP− proliferating cells in the trailing region behind the expanding vascular front (iv; EdU+/GFP−, blue arrowheads). Schematic summarizes the experimental protocol. BL, body length. (F,G) Whole-mount *Tg(−0.8flt1:RFP)*; *Tg(gata4:GFP)* ventricles from 25-mm (F) and 28-mm (G) -long zebrafish. Dotted lines outline the nonvascularized area (cyan) and lack of cortical CM (*gata4*:GFP+) coverage (red) at ventral ventricular base. Dashed boxes indicate regions shown at higher magnification to the right. AVC, atrioventricular canal; BA, bulbus arteriosus; V, ventricle. Scale bars: 100 µm.

in shaping the ventricular wall through cellular and molecular interactions. Our findings highlight both conserved and species-specific aspects of coronary vasculature development, provide new molecular tools for investigating coronary vessel identity and reveal how coronary vessel–CM interactions are modulated by angiogenic signaling levels.

This study reveals a dual-origin model of coronary vessel formation in zebrafish, identifying the emergence of coronary sprouts from both the BA and the AVC. Our data indicate that the earliest vessel arises from the BA, followed by a second vessel

sprouting from the AVC. Previous studies have shown that the coronary vasculature initiates from the AVC endothelium in zebrafish (Harrison et al., 2015), while in giant danio (*Devario malabaricus*), the coronary vasculature has a similar dual origin (Shifatu et al., 2018). Also, more recently Mizukami et al. showed in zebrafish that the hypobranchial artery expands caudally to reach the BA at 36 dpf (Mizukami et al., 2023), further supporting our observations. Interestingly, this pattern parallels the coronary vessel emergence in the human embryonic heart, where the early coronary plexus forms around the peritruncal region of the

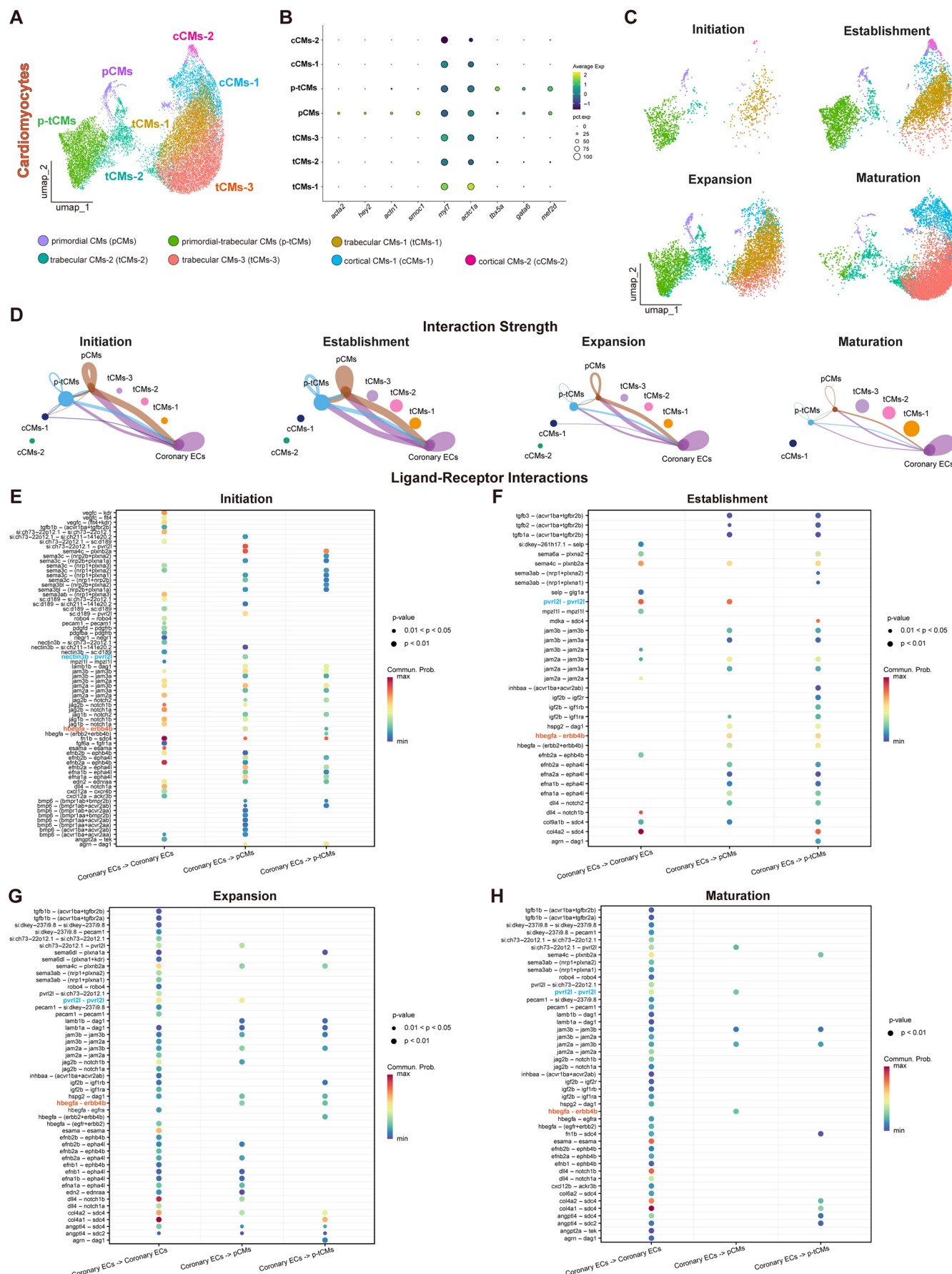

**Fig. 7.** See next page for legend.

**Fig. 7. Coronary vessel–CM molecular interactions during cardiac development.** (A-C) Combined UMAP plot (A) and split UMAP plots (C) of CMs visualizing seven subtypes identified across cardiac development, and dot plot showing average expression and abundance of selected marker genes corresponding to each subtype (B). (D) Circle plots showing differential cell–cell communication networks between coronary ECs and CM subtypes across cardiac development. Each node in the circle plot represents a cell cluster, with the node size being proportional to cell number. Edges connect nodes, indicating communication between cell clusters. Edge width represents communication strength. (E-H) Potential ligand–receptor interactions within coronary ECs, and between coronary ECs with primordial CMs (pCMs) and primordial-trabecular CMs (p-tCMs) across developmental stages of initiation (E), establishment (F), expansion (G) and maturation (H). *nectin3b-pvrl2l/pvrl2l-pvrl2l* and *hbegfa-erbb4b* interactions are highlighted in blue and orange text, respectively.

outflow tract (Bogers et al., 1989; Silva-Junior et al., 2009; Tomanek, 2005).

Based on anatomical landmarks and quantitative mapping, we have defined four stages of coronary network development: initiation, establishment, expansion and maturation. The coronary network expands in a dorsal-to-ventral and right-to-left direction, suggesting that either internal cardiac architecture or local vascular cues guide this process. These patterns of coronary vessel expansion are consistent with those observed in mammals (Chen et al., 2014; Red-Horse et al., 2010).

Despite growing interest in coronary vessel biology, the venous portion of the coronary network has remained uncharacterized owing to the lack of specific markers (Gancz et al., 2019; Harrison et al., 2015, 2019; Marín-Juez et al., 2016). Here, we developed a *TgBAC(sele:EGFP)* line that is specifically expressed in coronary venous ECs. Using this tool, we monitored coronary vein development and observed that venous vessels emerge early and expand progressively following the dorsal-to-ventral and right-to-left expansion pattern typical of coronary growth. Notably, *sele*: EGFP expression was undetectable in $-0.8flt1$:RFP$^{high}$ arterial vessels, allowing for the first time venous and arterial coronary vessels to be distinguished in zebrafish.

Further scRNA-seq analyses identified arterial, venous, capillary and lymphatic ECs. The relative abundance of these endothelial populations changed across developmental stages, with all subtypes showing progressive expansion from the initiation to the maturation stages, paralleling our *in vivo* data. In addition to known markers, our dataset identified new subtype-specific marker genes. Specifically, *tppp3*, *hpn* and *ano7* were enriched in capillaries and *gig2j* and *gjb10* in coronary veins.

Our data highlight the central role of Vegfa signaling in orchestrating multiple aspects of coronary development. Loss of *vegfaa* impairs LCA formation, EC proliferation, and spatial patterning, resulting in hypovascularization and disorganized vessel architecture. Conversely, increased Vegfa signaling via *flt1* deletion or *vegfaa* overexpression leads to regional hypervascularization, increased vessel density, and disrupted spatial organization. These phenotypes align with previous studies showing that Vegfa regulates angiogenesis in a dose-dependent manner, establishing vascular hierarchy and morphogenetic fidelity (Karra et al., 2018; Wang et al., 2024).

To further investigate how alterations in coronary network formation affect CM development, we analyzed CM growth in these Vegfa signaling gain- and loss-of function models. In *vegfaa* mutants, directional CM expansion was lost, and overall proliferation reduced. In *flt1* mutants, we observed enhanced CM coverage, consistent with observations in cardiac regeneration (Wang et al., 2024). Interestingly, stronger *vegfaa* overexpression reduced CM

coverage, suggesting that excessive Vegfaa decouples endothelial-CM developmental programs, as previously proposed during heart regeneration (Wang et al., 2024). It is worth noting that the coronary vasculature in these animals was strongly perturbed, raising the possibility that instructive angiocrine signals might also be impaired in these vessels. These data underscore the need for precisely tuned angiogenic signaling to maintain the coordinated growth of endothelial and myocardial tissues.

Finally, our scRNA-seq analyses identified stage-specific shifts in CM populations, including primordial, trabecular and cortical subtypes, recapitulating previous lineage-tracing analyses reporting late emergence of trabecular-derived cortical CMs (Gupta and Poss, 2012). We also observed a marked increase in cortical CMs throughout development, mirroring that of the coronary plexus, further supporting the coupling between cortical CM growth and coronary network expansion. Interestingly, the pCM and p-tCM subtypes showed the strongest interactions with coronary ECs, suggesting that developing coronaries might signal to pCMs and p-tCMs to support cortical CM expansion. In addition, ligand–receptor interaction analyses revealed signaling axes, such as *hbegfa–erbb4b* (paracrine cues) and *pvrl2l–pvrl2l* (cell–cell adhesion), known to regulate myocardial proliferation and endothelial behavior (Iwamoto and Mekada, 2006; Iwamoto et al., 2003; Kinugasa et al., 2012; Takai et al., 2003). These data support a model in which coronary ECs act as both instructive and structural components, actively shaping the cardiac microenvironment through temporally and spatially regulated signaling programs. The extent to which the interaction between coronaries and CMs is direct or mediated through intermediate factors, or involves both mechanisms, warrants further investigation.

## MATERIALS AND METHODS
### Zebrafish lines and manipulations

All experimental procedures involving zebrafish (*Danio rerio*) were conducted in strict adherence to institutional and national animal welfare regulations. In this study, zebrafish were raised in system water with 20 fish per 3-l tank.
 We used the previously published lines *TgBAC(etv2:EGFP)$^{ci1}$* (Proulx et al., 2010), *Tg(−0.8flt1:RFP)$^{hu5333}$* (Bussmann et al., 2010), *Tg(dll4:TagRFP)$^{sfc7}$* (Marín-Juez et al., 2016), *Tg(lyve1b:dsRed)$^{nz101}$* (Okuda et al., 2012), *Tg(fli1a:EGFP)$^{y1}$* (Lawson and Weinstein, 2002), *Tg(myl7:CreER)$^{pd10}$* (Kikuchi et al., 2010), *Tg(βactin2:loxP-mTagBFP-STOP-loxP-vegfaa)$^{pd262}$* (Karra et al., 2018), *Tg(gata4:GFP)$^{ae1}$* (Heicklen-Klein and Evans, 2004), *vegfaa$^{bns1}$* (Rossi et al., 2016), *flt1$^{bns29}$* (Matsuoka et al., 2016) and *scube2$^{as404}$* (Tsao et al., 2022). To induce recombination in juvenile fish, *Tg(myl7:CreER)*; *Tg(βact2:BS-vegfaa)* fish were bathed in 5 μM tamoxifen (4-OHT; Sigma-Aldrich, H7904) for 24 h in dark conditions as described previously (Karra et al., 2018) and put them back in system water after treatment.

### Generation of the *TgBAC(sele:EGFP)* zebrafish line
To generate *TgBAC(sele:EGFP)* fish, the BAC clone DKEY-51E6 containing *sele* was modified by Red/ET recombineering technology (Gene Bridges) as previously described (Bussmann and Schulte-Merker, 2011; Suster et al., 2011).

### Histological analyses
O-dianisidine staining was performed as described (Detrich et al., 1995). Ventricular section and whole-mount immunostaining were performed as previously described (Bakis et al., 2023; Marín-Juez et al., 2016). Primary antibodies used were: anti-GFP (Aves Labs, CGFP-1020; 1:500), anti-tRFP (Evrogen/Sapphire North America, AB233; 1:500), anti-Caveolin-1 (BD Transductions Laboratories, 610407; 1:100), anti-cTnT (Developmental Studies Hybridoma Bank, ct3; 1:100). Secondary antibodies used were Alexa Fluor 488 Goat Anti-Chicken IgG (H+L) (Thermo Fisher Scientific, A-11039; 1:750), Alexa Fluor 568 Goat Anti-Mouse IgG (H+L) (Thermo Fisher Scientific, A-11004; 1:750), Alexa Fluor 568 Goat Anti-Rabbit IgG

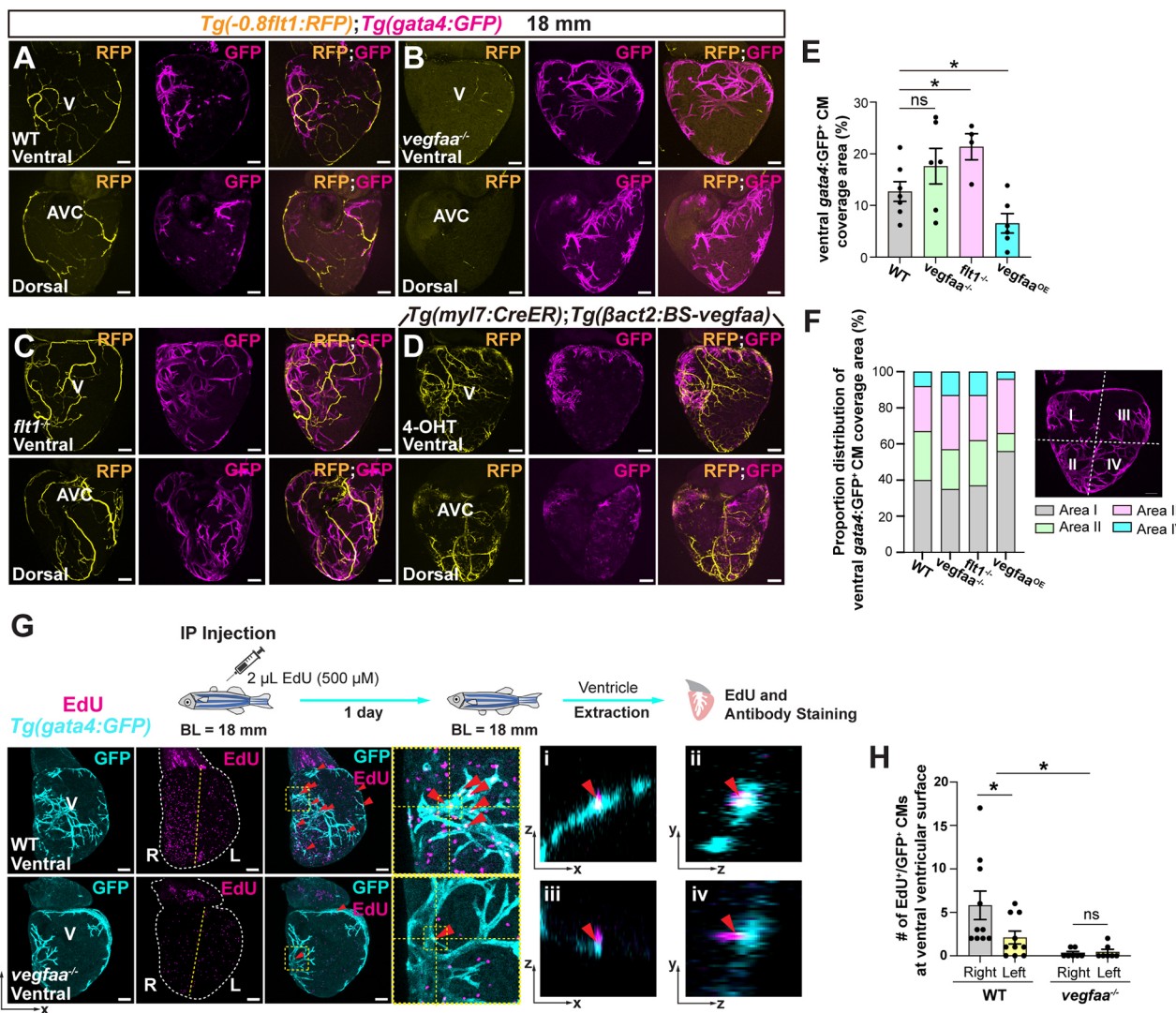

**Fig. 8. Manipulation of Vegfa signaling disrupts coronary–CM interactions.** (A-D) Whole-mount *Tg(−0.8flt1:RFP)*; *Tg(gata4:GFP)* ventricles from 18-mm-long WT (A), rescued *vegfaa⁻ᐟ⁻* (B), *flt1⁻ᐟ⁻* (C) and *Tg(myl7:creER)*; *Tg(βact2:BS-vegfaa)* treated with 4-OHT (*vegfaaᴼᴱ*) (D). (E,F) Percentage (E) and proportion distribution (F) of ventral *gata4*:GFP⁺ CM coverage area in 18-mm-long WT, rescued *vegfaa⁻ᐟ⁻*, *flt1⁻ᐟ⁻* and *Tg(myl7:creER)*; *Tg(βact2:BS-vegfaa)* treated with 4-OHT (*vegfaaᴼᴱ*). (G) Whole-mount *Tg(gata4:GFP)* ventricles from 18-mm-long WT and *vegfaa⁻ᐟ⁻* stained for GFP (growing cortical CMs, cyan) and EdU (proliferating cells, magenta). Yellow dashed lines divide the ventral ventricular surface into right (R) and left (L) sides. Red arrowheads point to EdU⁺/GFP⁺ cells. Dashed boxes indicate regions shown at higher magnification to the right. Orthogonal views of WT in the *xz* axis (i) and *yz* axis (ii) and *vegfaa⁻ᐟ⁻* in the *xz* axis (iii) and *yz* axis (iv) are shown. (H) Number of EdU⁺/GFP⁺ CMs on the right and left sides of the ventral ventricular surface in WT and rescued *vegfaa⁻ᐟ⁻* 18-mm-long zebrafish. Data in graphs are expressed as mean±s.e.m. ns, no significant difference. *P<0.05 (two-tailed, Mann–Whitney *U* test). Schematic summarizes the experimental protocol. BL, body length; IP, intraperitoneal; AVC, atrioventricular canal; V, ventricle. Scale bars: 100 µm.

(H+L) (Thermo Fisher Scientific, A-11036; 1:750) and Alexa Fluor 647 Goat Anti-Rabbit IgG (H+L) (Thermo Fisher Scientific, A-21244; 1:750). DAPI (4′,6-diamidino-2-phenylindole) was used as a nuclear counterstain. For HCR RNA-FISH (Molecular Instruments), the manufacturer's protocol for whole-mount zebrafish embryos and larvae was followed with modifications during sample preparation and fixation as previously described (Bakis et al., 2023). EdU staining was performed by using the Click-iT™ EdU Cell Proliferation Kit for Imaging with Alexa Fluor™ 555 dye (Invitrogen, C10338) following the manufacturer's instructions. EdU (2 µl, 500 µM) was injected intraperitoneally 1 day or 3 days before the extraction of the heart, depending on the experiment.

### Imaging and quantification

Anesthetized fish were dissected, and ventricles were extracted and immediately put in freshly prepared 1× PBS (Sigma-Aldrich, P4417) for at least 5 min and rinsed three times followed by 4% paraformaldehyde treatment for 5 min. Ventricles were then mounted on a glass-bottom dish

(ibidi, 80206) using 1.5% low melting agarose (Sigma-Aldrich, A4018). The ventricles were submerged in agarose without touching the bottom surface of the dish, and were later flipped to image both sides of ventricles. Imaging was performed on a Leica TCS SP8 laser scanning confocal microscope. O-dianisidine staining was imaged using a Nikon SMZ18 stereomicroscope. Quantifications were performed using ImageJ/Fiji from maximum projected confocal images. 3D reconstruction was performed using Imaris 10.2 (Oxford Instruments). Vascular coverage area quantifications were carried out using Tubeness (ImageJ/Fiji).

### Single-cell dissociation

One pool of ventricles per developmental stage – initiation (10 mm) 25 ventricles, establishment (14 mm) 18 ventricles, expansion (18 mm) 12 ventricles and maturation (28 mm) 5 ventricles – was prepared using WT zebrafish. Euthanized fish were dissected, and ventricles were extracted in calcium-, magnesium-free HBSS (Sigma-Aldrich, H9394) supplemented with 20 U/ml heparin (Sigma-Aldrich, H3393). Ventricles were washed in

fresh HBSS and transferred into 1 ml of 1 mg/ml of collagenase (Gibco, 17101015) dissolved in HBSS for approximately 30 min digestion on a thermal mixer at 32°C and 700 rpm. Ventricles were mechanically disrupted by pipetting every 10 min. Samples were centrifuged at 2000 rpm ($\sim$376 $g$) for 5 min at room temperature (RT) and supernatants discarded. Cells were digested with 1 ml 1× TrypLE Express Enzyme (Gibco, 12605010) for 15 min on a thermal mixer at 32°C and 700 rpm and pipetted every 5 min. Enzymatic digestion was stopped by adding 1 ml of 20% fetal bovine serum (Thermo Fisher Scientific, A3160701) in HBSS to extracts. Tissue extracts were strained through a 70 µm cell strainer and centrifuged at 2000 rpm ($\sim$376 $g$) for 5 min at RT. The supernatants were discarded, and pellets were washed with 1 ml of 20% fetal bovine serum in HBSS. After centrifugation at 2000 rpm ($\sim$376 $g$) for 5 min at RT, cells were resuspended in 80 µl of 20% fetal bovine serum in HBSS and strained through a 40 µm cell strainer. Cells were counted and viability (>85%) was assessed using 0.4% Trypan Blue (Sigma-Aldrich, T8154). Cells were kept on ice and immediately processed for scRNA-seq.

## scRNA-seq and analysis

For scRNA-seq, 14,000 cells were used for each sample and the Chromium Next GEM Single Cell 3′ Reagent Kit v3.1 (10x Genomics, PN-1000269) was used following the manufacturer's protocol. Each sample was indexed individually using the Dual Index Kit TT Set A (10x Genomics, PN-1000215). After quality control, libraries were run on a NovaSeq 6000 system (Illumina) at a depth of approximately 300 M reads/sample. FASTQ files were processed individually in 10x Genomics Cloud Analysis using the Cell Ranger Count v6.1.2 pipeline and reads were aligned to GRCz11 v4.3.2 reference genome (Lawson et al., 2020). Filtered gene expression matrices were imported and further processed in Seurat v4 (Hao et al., 2021). Samples were integrated to create a single aggregated Seurat object, and low-quality cells containing <200 unique features were removed. Batch effect correction was performed using the 'data integration' and 'ScTransform' functions in Seurat. To increase the retrieval of CMs characterized with high mitochondrial RNA content, an initial filtering was applied to remove cells with >40% mitochondrial RNA (Carey et al., 2024). After initial clustering and annotation of CMs, non-CM cells with >12% mitochondrial RNA were excluded from further analysis. The remaining cells were then re-clustered and annotated based on the expression of marker genes. Following normalization, data visualization was performed in Seurat, incorporating previously published R code with modifications (Carey et al., 2024). Quality control metrics are listed in Table S1. The number of cells retained at each filtering step is listed in Table S2. All marker genes in each cluster are listed in Table S3. The potential ligand–receptor interactions in each developmental stage are listed in Table S4 (initiation), Table S5 (establishment), Table S6 (expansion) and Table S7 (maturation).

## Statistical analysis

Data were statistically analyzed and graphics were created in GraphPad Prism v.10. When two groups were compared, comparative statistics were performed using two-tailed, unpaired Student's $t$-test or Mann–Whitney $U$-test for non-parametric tests. Data were considered significant at $P<0.05$. The number of samples are mentioned in the graphs and/or represented by dots in the graphs. The $P$-values are indicated in the figure legends. Data in graphs are expressed as mean±s.e.m.

## Acknowledgements
We thank all Marín-Juez lab members, and Gursimran Kaur Bajwa for helping with scRNA-seq preparation and analysis and Eliza Sassu and Simon Schnebert for helpful discussions and comments on the article. We thank Lara Feulner for helping with scRNA-seq analysis. We thank Jeroen Bakkers for providing the DKEY-51E6 BAC clone, Kenneth Poss for providing the Tg(βact2:BS-vegfaa) line and Stefan Materna for providing the dll4:TagRFP plasmid. We thank Marina Drits from the animal facility at Centre de Recherche Azrieli for assistance with zebrafish husbandry and Elke Kuester-Schoeck and Vanesa Jimenez-Amilburu from the imaging facility (Platform d'Imagerie Microscopique) at Centre de Recherche Azrieli for assistance with confocal imaging.

## Competing interests
The authors declare no competing or financial interests.

## Author contributions
Conceptualization: M.A.R., R.M.-J.; Data curation: M.A.R., R.M.-J.; Formal analysis: M.A.R., G.K.K., S.Z., S.L.V., S.M.K., R.M.-J.; Funding acquisition: R.M.-J.; Investigation: M.A.R., R.M.-J.; Methodology: M.A.R., G.K.K., S.Z., S.L.V., S.M.K., A.N.L., R.-B.Y., S.-L.L.; Project administration: R.M.-J.; Resources: M.A.R., S.Z., S.L.V., A.N.L., R.-B.Y., S.-L.L., R.M.-J.; Supervision: R.M.-J.; Validation: M.A.R., R.M.-J.; Visualization: M.A.R., R.M.-J.; Writing – original draft: M.A.R., R.M.-J.; Writing – review & editing: M.A.R., G.K.K., R.M.-J.

## Funding
M.A.R. was supported by a Merit Scholarship of Faculty of Medicine, Université de Montréal, and a FRQ Doctoral Training Scholarship from Fonds de recherche du Québec. G.K.K. and S.M.K. were supported by Fonds de recherche du Québec Postdoctoral Fellowships. R.M.-J. was supported by Fonds de recherche du Québec Junior-1 and Junior-2 awards. The Marín-Juez lab was supported by the Natural Sciences and Engineering Research Council of Canada (RGPIN-2021-03011) and the Canadian Institutes of Health Research (PJT-178037). Open Access funding provided by the Natural Sciences and Engineering Research Council of Canada and the Canadian Institutes of Health Research. Deposited in PMC for immediate release.

## Data and resource availability
Raw sequencing files are available on Gene Expression Omnibus database under accession number GSE298874. All other relevant data and details of resources can be found within the article and its supplementary information.

## Peer review history
The peer review history is available online at https://journals.biologists.com/dev/lookup/doi/10.1242/dev.205065.reviewer-comments.pdf

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
