## [Peer Review File · Development (Cambridge, England)]

Developmental single-cell atlas of coronary vessel growth and cardiomyocyte interaction in zebrafish

Muhammad Abdul Rouf, Gülsüm Kayman Kürekçi, Shaoqiu Zhang, Stéphanie Larrivée Vanier, Sarah M. Kamel, Ann Nee Lee, Ruey-Bing Yang, Shih-Lei Lai and Rubén Marín-Juez
DOI: 10.1242/dev.205065

Editor: Benoit Bruneau

Review timeline

Original submission:	3 July 2025
Editorial decision:	26 August 2025
First revision received:	24 November 2025
Accepted:	12 December 2025

Original submission

First decision letter

MS ID#: dev.205065

MS TITLE: Developmental single-cell atlas of coronary growth and cardiomyocyte interaction in zebrafish

AUTHORS: Muhammad Abdul Rouf, Gülsüm Kayman Kürekçi, Shaoqiu Zhang, Stéphanie Larrivée Vanier, Sarah M. Kamel, Ann Nee Lee, Ruey-Bing Yang, Shih-Lei Lai and Rubén Marín-Juez

Dear Dr Marín-Juez,

I have now received all the referees' reports on the above manuscript, and have reached a decision. The referees' comments are appended below, or you can access them online: please go to: *****

As you will see, the referees express considerable interest in your work, but have some significant criticisms and recommend a substantial revision of your manuscript before we can consider publication. If you are able to revise the manuscript along the lines suggested, which may involve further experiments, I will be happy to receive a revised version of the manuscript. Your revised paper will be re-reviewed by one or more of the original referees, and acceptance of your manuscript will depend on your addressing satisfactorily the reviewers' major concerns. Please also note that Development will normally permit only one round of major revision. If it would be helpful, you are welcome to contact us to discuss your revision in greater detail. Please send us a point-by-point response indicating your plans for addressing the referees' comments, and we will look over this and provide further guidance.

Please attend to all of the reviewers' comments and ensure that you clearly highlight all changes made in the revised manuscript. Please avoid using 'Tracked changes' in Word files as these are lost in PDF conversion. I should be grateful if you would also provide a point-by-point response detailing how you have dealt with the points raised by the reviewers in the 'Response to Reviewers' box. If you do not agree with any of their criticisms or suggestions please explain clearly why this is so.

Reviewer 1*Advance summary and potential significance to field*

The manuscript by Rouf and colleagues presents a comprehensive analysis of coronary vessel development in zebrafish. Using high resolution microscopy, the authors delineate key stages of the coronary vasculature development, encompassing initiation, establishment, expansion, and maturation. Single cell transcriptomics analyses revealed previously unrecognized coronary endothelial cell (EC) subpopulations and novel markers of these cell types. Their analyses further uncovered signaling crosstalks between coronary ECs and cardiomyocytes, highlighting how coronary vessels act as scaffolds that guide cardiomyocyte expansion.

The manuscript has several major strengths. By leveraging the advantages of the zebrafish model, the study provides a novel and comprehensive developmental atlas of the coronary vasculature, addressing a long-standing gap in the field despite the clinical importance of these vessels. The authors employ high-resolution, time course microscopy analyses to clearly capture each developmental stage, enabling accurate spatial and temporal interpretation of coronary vessel formation. The single-cell RNA-seq analysis is a valuable resource, revealing new marker genes of the coronary endothelial cell subpopulations. Additionally, the generation of the TgBAC(sele:EGFP) transgenic line that selectively labels venous coronary ECs is an important tool for future studies. Collectively, these resources would enable more accurate analysis of various cell populations. Lastly, by integrating EC and cardiomyocyte transcriptomic datasets, the study offers unique insights into the reciprocal developmental interactions between these two lineages, enhancing our understanding of their coordinated growth.

Comments for the author

1. Standard quality control metrics of the single cell transcriptomics data should be provided that would enable readers to better evaluate and interpret the dataset. Specifically, it would be helpful to include the number of cells that passed each filtering step and were retained for the final analysis. Additionally, information on the number of cells within each identified cluster would clarify the representation of different cell populations. Finally, information on the median or mean number of reads and genes per cell across the various clusters, ideally accompanied by distribution plots, would offer valuable insight into the depth and quality of the sequencing data.
2. The single-cell analysis presented in the study is a valuable resource that supports many of the authors' observations. However, this component currently feels somewhat disconnected from the rest of the manuscript, and several opportunities to more fully integrate it into the functional and developmental analyses appear to have been missed.
 - a. While the authors successfully identified novel marker genes for coronary endothelial cell subpopulations, these markers were not utilized in their functional analyses of *vegfaa* and *scube2*. For example, in the *vegfaa* overexpression experiment, the arterial marker *flt1* was found to be highly induced as a result of the overexpression, but no attempt was made to distinguish arterial from venous ECs using the newly identified markers. Incorporating these markers could provide a more nuanced interpretation of how specific EC subtypes respond to the various perturbations. Could the authors clarify whether this analysis could be performed using the newly identified markers, or to explain the rationale for not including it?
 - b. The authors nicely show that *gata4:EGFP+* cardiomyocytes emerge concurrently with developing coronary vessels during the expansion stage. Their scRNA-seq data also allowed them to distinguish several cardiomyocyte subpopulations which vary in abundance across developmental stages. Analyzing proliferation levels within these subpopulations by examining the expression of cell cycle-related genes could provide further insights into how coronary vessel development influences cardiomyocyte expansion.
 - c. It would be informative to assess the expression patterns of VEGF receptors across the distinct coronary EC subpopulations identified in the single-cell dataset, especially given that VEGF receptor expression is known to vary between EC subtypes. Such analysis could clarify which populations are most likely to respond directly to VEGFA, thereby strengthening the interpretation

of the vegfaa functional analyses and deepening our understanding of how VEGFA signaling influences coronary vessel patterning.

d. To enhance the utility of the dataset for the broader research community, it would be helpful to provide the full analysis output. Specifically, a supplementary table listing all marker genes for each cell cluster should be included. Additionally, a table detailing all signaling interactions identified in the study would further support the interaction maps depicted in Figures 7D and S10.

Reviewer 2

Advance summary and potential significance to field

The work by Rouf and colleagues addresses the role of the developing zebrafish coronary system in the formation of the ventricular wall – in particular in the rise and expansion of the cortical cardiomyocyte layer—. Furthermore, the authors characterize the diversity of endothelial cells during coronary vascular formation to propose a roadmap for zebrafish coronary morphogenesis. Although the subject of this work is relevant to the cardiovascular development field and the authors have selected an appropriate set of techniques to research on the topic, some of the conclusions drawn by the authors are not supported by their results. Therefore, additional experimentation is required to complete the manuscript and render it suitable for publication in Development.

Comments for the author

I have enjoyed reading the manuscript, which includes a significant amount of interesting data. However, I am less fond of the way the results are interpreted or speculations on such results are presented. This is already shown in the misleading title (sc-RNAseq-based EC mapping can seldom reveal a functional coordination between cell types, it can suggest it. Then, functional analyses – gain-of- or loss-of-function– will confirm the point).

My specific comments are as follows:

1. The abstract should be more open and consider that coronary vessels participate in the shaping of the ventricular wall, but it is known that coronary cells are not the only ones involved in patterning the ventricular myocardium (consider the role of Notch-dependent endocardial signalling).
2. Using anatomical terms as based on analogy is extremely dangerous. I strongly discourage the use of LAD to refer to the vessel described between lines (l) 97 and 102. LAD (mammals) is a denomination closely associated to the four-chambered heart and has a septal component to it (in most micromammals the LAD artery runs along the interventricular septum). Moreover, in the zebrafish, the artery does not "descend". Thus, in the absence of four chambers, a septum and given the anatomical arrangement of the zebrafish heart, the use of LAD is not appropriate.
3. The description of anatomical entities should be precise and avoid by all means to confuse the reader. For example, data on the expansion of the hypobranchial artery to the bulbus arteriosus or the formation of lymphatic vessels in the same area). This needs to be considered and discussed as related with the connection of the arterial coronary network with the corresponding efferent vessel supplying oxygenated blood to the system. Then, how and where is the coronary vein drainage established? Finally, how does the initiation of blood circulation in the nascent coronary network affect to the proliferative expansion of the vessels.
4. Part of the results illustrated in Fig.1 overlap with previous work by other authors (Harrison et al., 2015, Dev Cell 33:442). Then, when additional, novel results are shown they do not receive the attention they deserve. For example, in the description in l-105 of a dorsal EGFP+/RFP-LOW vessels close to the so-called LAD...What kind of vessel is this and why its proximity to the "LAD" is relevant?. In l.181 "spatially separated" needs to be detailed (e.g., is one vessel dorsal to the other?)

5. Splitting channels in the illustrating images is important to assess multiple stainings/fluorescence, most especially when one of the markers displays a faint (LOW) expression (e.g. Fig1F-I shows the channels splitted whereas Fig.1R-U and W-Y have the channels merged). This should be considered throughout the whole paper.
6. The authors need to attain sufficient cell resolution when dealing with EC-CM interactions; so far, their whole mount stainings do not suffice to support their conclusions.
7. Regarding cell proliferation (l.270) Vegf and Shh form part of a Retinoic Acid (RA)-dependent signalling network controlling endothelial proliferation in vertebrates (Lai et al., 2003, Development 130:6465; Bohnsack et al., 2004, Genes Dev 18:1345). RA ought to be studied by the authors in the context of zebrafish coronary vessel formation.
8. Some conclusions such as "Colectively...(l.331)" are generic. That altered VEGF expression impacts vascular development has been well-known for decades (Drake and Little, 1999, J Histochem Cytochem 47:1351). Please, be more specific in the interpretation of the findings.
9. The emergence of cortical CMs in relation to the expansion of the early coronary network (l.339) requires the analysis of cell polarisation/orientation along the process.
10. The results from the secondary subclustering analyses performed to further resolve the heterogeneity among ECs (l.216-219) are not properly detailed. What is the meaning of the findings shown in Fig.S5B-S5G?
11. The stainings provided for the genes listed in l. 228 (tppp3, hpn, ano7, gig2j and gjb10, see Fig.3J-O) are extremely poor and difficult to believe as shown.
12. The role of the nascent coronary vasculature in scaffolding cortical CM organization is shown indirectly only. By identifying genes related to cell:cell adhesion from their sc-RNAseq data the authors have an excellent tool to spot genes potentially involved in a EC-CM interaction mechanism, but the mere identification does not confirm anything –the functional assessment of the putative role assigned to each molecule has to be completed. Most likely, alternative strategies will be needed to test whether the scaffolding role of the coronary vasculature is physical (the contact with the vessels support cortical CM development), instructive (coronary ECs secrete relevant developmental cues to the myocardium), or both.

First revision

Author response to reviewers' comments

Reviewer 1: SUMMARY OF THE ADVANCE MADE IN THIS PAPER AND ITS POTENTIAL SIGNIFICANCE TO THE FIELD

The manuscript by Rouf and colleagues presents a comprehensive analysis of coronary vessel development in zebrafish. Using high resolution microscopy, the authors delineate key stages of the coronary vasculature development, encompassing initiation, establishment, expansion, and maturation. Single cell transcriptomics analyses revealed previously unrecognized coronary endothelial cell (EC) subpopulations and novel markers of these cell types. Their analyses further uncovered signaling crosstalks between coronary ECs and cardiomyocytes, highlighting how coronary vessels act as scaffolds that guide cardiomyocyte expansion.

The manuscript has several major strengths. By leveraging the advantages of the zebrafish model, the study provides a novel and comprehensive developmental atlas of the coronary vasculature, addressing a long-standing gap in the field despite the clinical importance of these vessels. The authors employ high-resolution, time course microscopy analyses to clearly capture each

developmental stage, enabling accurate spatial and temporal interpretation of coronary vessel formation. The single-cell RNA-seq analysis is a valuable resource, revealing new marker genes of the coronary endothelial cell subpopulations. Additionally, the generation of the *TgBAC(sele:EGFP)* transgenic line that selectively labels venous coronary ECs is an important tool for future studies. Collectively, these resources would enable more accurate analysis of various cell populations. Lastly, by integrating EC and cardiomyocyte transcriptomic datasets, the study offers unique insights into the reciprocal developmental interactions between these two lineages, enhancing our understanding of their coordinated growth.

We thank the reviewer for their supportive and constructive comments.

SUGGESTIONS TO AUTHORS

1. Standard quality control metrics of the single cell transcriptomics data should be provided that would enable readers to better evaluate and interpret the dataset. Specifically, it would be helpful to include the number of cells that passed each filtering step and were retained for the final analysis. Additionally, information on the number of cells within each identified cluster would clarify the representation of different cell populations. Finally, information on the median or mean number of reads and genes per cell across the various clusters, ideally accompanied by distribution plots, would offer valuable insight into the depth and quality of the sequencing data.

We have now included a summary of the standard quality control metrics for our single-cell RNA-seq dataset. We now report the number of cells retained after each filtering step (Supplementary Table 2 and Materials and Methods-Single-cell RNA sequencing and analysis), the median and mean number of genes and reads per cell (Supplementary Table 1 and Materials and Methods-Single-cell RNA sequencing and analysis) accompanied by violin plots to visualize these metrics across stages, and the number of cells in each identified cluster (Supplementary Figure 6). We believe these additions make the dataset easier for readers to evaluate and interpret.

2. The single-cell analysis presented in the study is a valuable resource that supports many of the authors' observations. However, this component currently feels somewhat disconnected from the rest of the manuscript, and several opportunities to more fully integrate it into the functional and developmental analyses appear to have been missed.

a. While the authors successfully identified novel marker genes for coronary endothelial cell subpopulations, these markers were not utilized in their functional analyses of *vegfaa* and *scube2*. For example, in the *vegfaa* overexpression experiment, the arterial marker *flt1* was found to be highly induced as a result of the overexpression, but no attempt was made to distinguish arterial from venous ECs using the newly identified markers. Incorporating these markers could provide a more nuanced interpretation of how specific EC subtypes respond to the various perturbations. Could the authors clarify whether this analysis could be performed using the newly identified markers, or to explain the rationale for not including it?

Thanks for this suggestion. We agree that assessing *vegfaa* perturbation effects on distinct coronary EC subtypes using the newly identified markers would provide valuable information.

To address this point, we performed *in situ* HCR coupled with immunostaining in *vegfaa*^{-/-} (Figure A) and *vegfaa*^{OE} (Figure B) *TgBAC(etv2:EGFP); Tg(-0.8flt1:RFP)* ventricles. Despite multiple attempts, the expression patterns of these markers were variable, preventing us from drawing any conclusions about how EC subtypes respond to *Vegfa* signaling perturbations. Notably, some of these markers are expressed at low levels in specific cell types, making reliable quantification extremely challenging. As mentioned by the reviewer, *Vegfaa* overexpression results in strong and ectopic activation of the *-0.8flt1:RFP* transgene. Our data indicate that this change does not reflect alterations in specification. Similarly, we observed strong ectopic expression of several of the newly identified markers (Figure B) upon *vegfaa* overexpression. To maintain clarity and data robustness, we decided not to include these preliminary data in the revised version.

Figure A. Wholemount *vegfaa*^{-/-} *Tg(etv2:EGFP); Tg(-0.8flt1:RFP)* zebrafish ventricles showing *tpp3*, *hpn*, *ano7*, *gig2j* and *gjb10* expression by *in situ* HCR

Figure B. Wholemount *Tg(myf7:CreER); Tg(Bact2:BS-vegfaa); Tg(etv2:EGFP); Tg(-0.8flt1:RFP)* treated with 4-OHT (*vegfaa*^{OE}) zebrafish ventricles showing *tpp3*, *hpn*, *ano7*, *gig2j* and *gjb10* expression by *in situ* HCR

b. The authors nicely show that *gata4:EGFP*⁺ cardiomyocytes emerge concurrently with developing coronary vessels during the expansion stage. Their scRNA-seq data also allowed them to distinguish several cardiomyocyte subpopulations which vary in abundance across developmental stages. Analyzing proliferation levels within these subpopulations by examining the expression of cell cycle-related genes could provide further insights into how coronary vessel development influences cardiomyocyte expansion.

We thank the reviewer for this suggestion. To assess proliferation within CM subtypes, we examined the expression of well-established cell cycle-related genes, including *mki67*, *pcna*, *aurka*, *aurkb*, *ccnb1* and *mcm5* across all CM subtypes identified in our scRNA-seq datasets (Supplementary Figure 15).

Based on these data, we observed that the majority of these proliferation markers, notably *mki67*, *pcna*, *aurka* and *mcm5* were enriched from initiation to expansion stages, particularly in primordial and primordial-trabecular CMs. These results support our data that CM

proliferation peaks during the expansion stage of coronary development. These new data are now discussed in the manuscript (L427-430).

c. It would be informative to assess the expression patterns of VEGF receptors across the distinct coronary EC subpopulations identified in the single-cell dataset, especially given that VEGF receptor expression is known to vary between EC subtypes. Such analysis could clarify which populations are most likely to respond directly to VEGFA, thereby strengthening the interpretation of the vegfaa functional analyses and deepening our understanding of how VEGFA signaling influences coronary vessel patterning.

We have now reanalyzed our scRNA-seq datasets and examined the expression of Vegf receptors (*flt1*, *kdrl*, *kdr* and *flt4*) across coronary EC subtypes throughout development (Supplementary Figure 11). We find that *flt1* and *kdrl* are broadly expressed across coronary EC subtypes but showing reduced expression in lymphatic ECs, *kdr* is expressed across all subtypes including lymphatics, and *flt4* expression is largely restricted to lymphatic ECs. These new data are now discussed in the manuscript (L332-335).

d. To enhance the utility of the dataset for the broader research community, it would be helpful to provide the full analysis output. Specifically, a supplementary table listing all marker genes for each cell cluster should be included. Additionally, a table detailing all signaling interactions identified in the study would further support the interaction maps depicted in Figures 7D and S10.

We have now included five new supplementary tables:

Supplementary Table 3: List of all marker genes identified for each cell cluster, along with associated statistics (average log fold change, p-value, adjusted p-value).

Supplementary Table 4-7: Summarizes all predicted ligand-receptor interactions between cell types identified in our analysis across developmental stages (initiation-Supplementary Table 4, establishment-Supplementary Table 5, expansion-Supplementary Table 6, maturation- Supplementary Table 7), which support the interaction maps shown in Figures 7D and S10 (Supplementary Figure 14 in the revised manuscript).

Reviewer 2: SUMMARY OF THE ADVANCE MADE IN THIS PAPER AND ITS POTENTIAL SIGNIFICANCE TO THE FIELD

The work by Rouf and colleagues addresses the role of the developing zebrafish coronary system in the formation of the ventricular wall –in particular in the rise and expansion of the cortical cardiomyocyte layer–. Furthermore, the authors characterize the diversity of endothelial cells during coronary vascular formation to propose a roadmap for zebrafish coronary morphogenesis. Although the subject of this work is relevant to the cardiovascular development field and the authors have selected an appropriate set of techniques to research on the topic, some of the conclusions drawn by the authors are not supported by their results. Therefore, additional experimentation is required to complete the manuscript and render it suitable for publication in Development.

We thank the reviewer for their supportive and constructive comments.

SUGGESTIONS TO AUTHORS

I have enjoyed reading the manuscript, which includes a significant amount of interesting data. However, I am less fond of the way the results are interpreted or speculations on such results are presented. This is already shown in the misleading title (sc-RNAseq-based EC mapping can seldom reveal a functional coordination between cell types, it can suggest it. Then, functional analyses –gain- of- or loss-of-function– will confirm the point).

The revised title is now: **Developmental single-cell atlas of coronary growth and cardiomyocyte interaction in zebrafish**

My specific comments are as follows:

1. The abstract should be more open and consider that coronary vessels participate in the shaping of the ventricular wall, but it is known that coronary cells are not the only ones involved in patterning the ventricular myocardium (consider the role of Notch-dependent endocardial signalling).

We have now modified the abstract to acknowledge that cardiac morphogenesis is regulated by different cell types and molecular cues.

2. Using anatomical terms as based on analogy is extremely dangerous. I strongly discourage the use of LAD to refer to the vessel described between lines (l) 97 and 102. LAD (mammals) is a denomination closely associated to the four-chambered heart and has a septal component to it (in most micromammals the LAD artery runs along the interventricular septum). Moreover, in the zebrafish, the artery does not "descend". Thus, in the absence of four chambers, a septum and given the anatomical arrangement of the zebrafish heart, the use of LAD is not appropriate.

We now refer to this vessel as the left coronary artery (LCA) to reflect its localization along the left ventricular curvature and arterial identity. We have updated the text and figures accordingly.

3a. The description of anatomical entities should be precise and avoid by all means to confuse the reader. For example, data on the expansion of the hypobranchial artery to the bulbus arteriosus or the formation of lymphatic vessels in the same area). This needs to be considered and discussed as related with the connection of the arterial coronary network with the corresponding efferent vessel supplying oxygenated blood to the system.

Based on our imaging analyses and those from others (PMID: 31702554; PMID: 31702553), two distinct endothelial populations coexist on the BA during early coronary vessel formation. Using the *TgBAC(etv2:EGFP)* line, we observed the first coronary vessel on the BA at 8 mm of body length, consistent with the previously reported hypobranchial artery extension onto the BA (PMID: 37605519). Since *etv2* marks endothelial progenitors rather than differentiated arterial or venous cells, we interpret these as coronary endothelial progenitors initiating coronary vasculature formation.

In parallel, using *TgBAC(sele:EGFP)* and *Tg(lyve1b:dsRed)* double transgenic animals, we identified a population of *sele:EGFP⁺/lyve1b:dsRed⁺* lymphatic endothelial cells on the BA at similar stages. These lymphatic cells appear on the BA surface and later extend toward the ventricle, where they lose *sele:EGFP* expression but maintain *lyve1b:dsRed* signal, consistent with their lymphatic identity and the previously reported timing of lymphatic development (PMID: 31702554; PMID: 31702553). We have now revised the manuscript to further clarify these points (L93-95 and L195-204).

3b. Then, how and where is the coronary vein drainage established?

This indeed is an interesting question that remains to be answered in zebrafish. Despite our previous efforts to monitor adult cardiac circulation with specific transgenic lines (e.g., *gata1* lines), these lines are silenced in adults.

3c. Finally, how does the initiation of blood circulation in the nascent coronary network affect to the proliferative expansion of the vessels.

We have now performed EdU incorporation assays combined with antibody staining using the *TgBAC(etv2:EGFP)* line to label coronary ECs before (16 mm) and after (19 mm) the onset of coronary perfusion. We find that the number of EdU⁺ coronary endothelial cells was significantly higher at 19 mm compared to 16 mm. These new data are now included in Supplementary Figure 4 and discussed in the manuscript (L154-159).

4a. Part of the results illustrated in Fig.1 overlap with previous work by other authors (Harrison et al., 2015, Dev Cell 33:442). Then, when additional, novel results are shown they do not receive the attention they deserve. For example, in the description in l-105 of a dorsal EGFP⁺/RFP-LOW

vessels close to the so-called LAD...What kind of vessel is this and why its proximity to the "LAD" is relevant?

We have now rewritten several sections of the manuscript to further emphasize our data. As part of our findings, we report the identification of a dorsal EGFP⁺/RFP^{low} vessel that expands from the ventricular base and is in close proximity to the LCA. We have now better described this vessel and its localization (L112-115).

With the available tools and knowledge of the coronary vasculature in zebrafish, it would be premature to speculate the possible relevance of this vessel.

4b. In l.181 "spatially separated" needs to be detailed (e.g., is one vessel dorsal to the other?)

We have now revised this section of the manuscript in more detail (L188-193).

5. Splitting channels in the illustrating images is important to assess multiple stainings/fluorescence, most especially when one of the markers displays a faint (LOW) expression (e.g. Fig1F-I shows the channels splitted whereas Fig.1R-U and W-Y have the channels merged). This should be considered throughout the whole paper.

We have now extensively restructured different figures throughout the manuscript to facilitate data assessment. In the case of Figures 1K-1N, 1R-1U and 1W-1Y, we have now separated the channels of these images and provided them in the Supplementary Figure 2.

6. The authors need to attain sufficient cell resolution when dealing with EC-CM interactions; so far, their whole mount stainings do not suffice to support their conclusions.

We have now performed high-resolution confocal z-stack imaging and generated new 3D reconstructions using the Imaris software (Figure 6B,B'). These analyses provide improved spatial resolution and more clearly depict the spatial relationship between coronary vessels and CM during the expansion 1 stage. These 3D reconstructions highlight sites of close EC-CM apposition, allowing visualization of their intimate structural association during development. We believe that these new data, together with analyses included in the revised manuscript (e.g., Figure 6B,C and new Figure 6D) and previously published data showing EC-CM interaction during development in whole mounts and ventricular sections (PMID: 31743664), further reinforce the notion of developmental EC-CM scaffolding and interactions.

Moreover, quantification of the physical distance between developing ECs and CMs using specific transgenic lines and confocal microscopy showed that the vast majority of these cells are in very close proximity (0-15 μ m). Importantly, we are aware that these data do not demonstrate direct cell-to-cell contact and we carefully revised the text in the manuscript to reflect this distinction.

7. Regarding cell proliferation (l.270) Vegf and Shh form part of a Retinoic Acid (RA)-dependent signalling network controlling endothelial proliferation in vertebrates (Lai et al., 2003, Development 130:6465; Bohnsack et al., 2004, Genes Dev 18:1345). RA ought to be studied by the authors in the context of zebrafish coronary vessel formation.

We have now examined the expression of RA-related genes across developmental stages in our scRNA-seq datasets, and these new data are included in Supplementary Figure 9 and discussed in the manuscript (L290-296).

8. Some conclusions such as "Collectively...(l.331)" are generic. That altered VEGF expression impacts vascular development has been well-known for decades (Drake and Little, 1999, J Histochem Cytochem 47:1351). Please, be more specific in the interpretation of the findings.

We have now revised the text to be more specific in the interpretation of our findings (L336-338).

9. The emergence of cortical CMs in relation to the expansion of the early coronary network (l.339) requires the analysis of cell polarisation/orientation along the process.

We have now analyzed CM sarcomere orientation and organization in relation to coronary vessel development. We performed cardiac troponin T (cTnT) staining in *TgBAC(etv2:EGFP); Tg(gata4:GFP)* ventricles, labeling coronary ECs and growing CM, respectively at the expansion stage (Figure 6D). These new data show that CMs trailing behind the expanding front display defined sarcomeres arranged perpendicularly near to coronaries, indicative of polarization and structural maturation. Furthermore, these CMs downregulate *gata4:GFP*, consistent with their transition toward a mature, more quiescent state.

In contrast, CMs at the expanding front are in close proximity to the coronary sprouting front and exhibit poorly defined sarcomeres while remaining *gata4:GFP*^{high}. These new data are discussed in the revised manuscript (L359-364).

10a. The results from the secondary subclustering analyses performed to further resolve the heterogeneity among ECs (l.216-219) are not properly detailed.

We have now elaborated this section of the manuscript in greater detail (L225-230).

10b. What is the meaning of the findings shown in Fig.S5B-S5G?

To describe other relevant cell clusters including EPDCs and mural cells 2 that might contribute to coronary development. These analyses could be a valuable resource of cell types and candidate genes for future studies investigating the mechanisms underlying coronary vessel formation and maturation. We have now rewritten this section in the manuscript (L230-231).

11. The stainings provided for the genes listed in l. 228 (*tppp3*, *hpn*, *ano7*, *gig2j* and *gjb10*, see Fig.3J- O) are extremely poor and difficult to believe as shown.

We have now performed new *in situ* HCR stainings to improve the quality of these data (Figure 3). We believe that these new data more convincingly support the expression of marker genes in coronary vessels. The expression of some of these markers is low and has only been possible to detect now thanks to advances in scRNA-seq technologies. In some cases, further increasing signal intensity compromised background levels and thereby reduced signal specificity.

In the case of *hpn* (Figure C), we were unable to improve staining conditions to more clearly show EC expression and therefore it has been removed from the manuscript

Figure C. Wholemount *Tg(etv2:EGFP); Tg(-0.8flt1:RFP)* zebrafish ventricles showing *hpn* expression by *in situ* HCR

12. The role of the nascent coronary vasculature in scaffolding cortical CM organization is shown indirectly only. By identifying genes related to cell:cell adhesion from their sc-RNAseq data the authors have an excellent tool to spot genes potentially involved in a EC-CM interaction mechanism, but the mere identification does not confirm anything –the functional assessment of the putative role assigned to each molecule has to be completed. Most likely, alternative strategies will be needed to test whether the scaffolding role of the coronary vasculature is physical (the contact with the vessels support cortical CM development), instructive (coronary ECs secrete relevant developmental cues to the myocardium), or both.

Our analyses identify pathways potentially mediating EC-CM interaction/communication. Among these candidates, genes related to cell adhesion (*nectin/pvrl2l*) and paracrine signaling (*hbegfa-erbb4b*) were prominently expressed at stages of vigorous coronary/CM expansion. Moreover, using different genetic models to manipulate Vegfa signaling, we were able to experimentally uncouple coronary-CM interactions. In addition, we previously reported that during regeneration coronary scaffolding is also essential for CM replenishment (PMID:31743664). In these studies, we provide evidence supporting both physical and instructive EC-CM interactions during both regeneration and development. While we agree with the reviewer that this is an important aspect of the interaction between coronaries and CMs, we believe that the data presented here, together with our previous work, provide sufficient evidence to reasonably support coronary scaffolding and coronary-CM interactions. Whether these interactions are by direct cell-cell contact or alternative mechanisms remains to be determined in future studies. Lastly, we believe that functional assessment of each molecule identified in our scRNAseq datasets potentially related to cell-cell adhesion is beyond the scope of this resource article.

Second decision letter

MS ID#: dev.205065R1

MS TITLE: Developmental single-cell atlas of coronary growth and cardiomyocyte interaction in zebrafish

AUTHORS: Muhammad Abdul Rouf, Gülsüm Kayman Kürekçi, Shaoqiu Zhang, Stéphanie Larrivée Vanier, Sarah M. Kamel, Ann Nee Lee, Ruey-Bing Yang, Shih-Lei Lai and Rubén Marín-Juez

Dear Dr Marín-Juez,

I am happy to tell you that your manuscript has been accepted for publication in Development, pending our standard publication integrity checks.

Reviewer 1

Advance summary and potential significance to field

The manuscript has been greatly improved by the inclusion of quality parameters for the scRNA-seq data as well as the additional analyses that better link this dataset to the biological observations.

Comments for the author

The authors have addressed all my concerns. I have only one minor suggestion regarding the description for Figure S6 panel B-C. It would be helpful if the authors could provide a more descriptive legend for the quality metrics (i.e. what parameters each graph represents) for the benefit of non-specialist readers.

Reviewer 2

Advance summary and potential significance to field

The authors have properly addressed all my comments on the first version of the manuscript. I appreciate the work done to provide new data, clarify complex aspects of the study, and avoid unnecessary speculation while interpreting the findings reported. On a final note, regarding point 12 of the answers provided by the authors to my comments, I understand the rationale of the work and acknowledge there is a strong set of evidences supporting CM-coronary interactions throughout

embryonic development (something in a way logical and in line with previously published papers, see: doi: 10.1161/CIRCRESAHA.107.160861). I do, however, recommend the authors to critically consider the use of the 'scaffolding' concept: a scaffold is a structure to support something without necessarily providing instructive cues to the supported matter (unless properly proved). Accordingly, I would kindly suggest to tune down the use of the term. Finally, I did not intend the authors to provide experimental data on all the sc-RNAseq candidate genes suspected to be involved in CM-coronary interactions –that, I understand, is well beyond the scope of the study– but it is obvious that having at least a candidate gene and some LOF data on such gene would have strengthen the CM 'scaffolding' phenomenon in the sense of active guidance of coronary blood vessel morphogenesis.